# Hierarchical Successor Representation for Robust Transfer

Changmin Yu [1]    Máté Lengyel [1 2]

## Abstract

The successor representation (SR) provides a powerful framework for decoupling predictive dynamics from rewards, enabling rapid generalisation across reward configurations. However, the classical SR is limited by its inherent policy dependence: policies change due to ongoing learning, environmental non-stationarities, and changes in task demands, making established predictive representations obsolete. Furthermore, in topologically complex environments, SRs suffer from spectral diffusion, leading to dense and overlapping features that scale poorly. Here we propose the Hierarchical Successor Representation (HSR) for overcoming these limitations. By incorporating temporal abstractions into the construction of predictive representations, HSR learns stable state features which are robust to task-induced policy changes. Applying non-negative matrix factorisation (NMF) to the HSR yields a sparse, low-rank state representation that facilitates highly sample-efficient transfer to novel tasks in multi-compartmental environments. Further analysis reveals that HSR-NMF discovers interpretable topological structures, providing a policy-agnostic hierarchical map that effectively bridges model-free optimality and model-based flexibility. Beyond providing a useful basis for task-transfer, we show that HSR's temporally extended predictive structure can also be leveraged to drive efficient exploration, effectively scaling to large, procedurally generated environments.

## 1. Introduction

Despite significant progress in reinforcement learning (RL) over the past two decades (Sutton & Barto, 1998; Matsuo

[1]Computational and Biological Learning Lab, Department of Engineering, University of Cambridge, Cambridge, United Kingdom [2]Department of Cognitive Science, Central European University, Budapest, Hungary. Correspondence to: Changmin Yu <changmin.yu98@gmail.com>.

*Proceedings of the 43rd International Conference on Machine Learning*, Seoul, South Korea. PMLR 306, 2026. Copyright 2026 by the author(s).

et al., 2022), achieving robust and efficient generalisation remains a major challenge (Cobbe et al., 2019). For instance, an agent navigating in a maze may encounter changes in goal locations or reward configurations while the environmental geometry remains unchanged. In such settings, the key to efficient learning is to construct robust, reusable state representations that capture persistent structure of the environment (Zhang et al., 2020; Agarwal et al., 2021), allowing for rapid adaptation when goals change. However, achieving robust transfer following task changes is difficult since the policy that is optimal for one task may be suboptimal for another, and associated policy adaptation could invalidate representations learned under the previous behaviour.

The successor representation (SR) framework offers a promising avenue for such transfer (Dayan, 1993). By decomposing value function into a linear combination of expected state occupancy and state-dependent reward function, the SR enables generalisation through rapid policy re-evaluation when the reward function changes. However, two bottlenecks limit the practical scope of SRs in transfer-oriented regimes. First, SRs are inherently policy dependent, since the predictive occupancy changes when the policy is updated. This issue becomes acute when generalisation requires not only re-evaluating state values but also re-optimising behaviour. Second, SR features are dominated by diffusion, leading to globally supported, overlapping features, a phenomenon particularly evident in spectral components of SRs (Mahadevan & Maggioni, 2007; Stachenfeld et al., 2017). The resultant state features lack interpretability and scale poorly as the environment becomes larger or more topologically complex (e.g., compartmentalised; Figure 1b). Consequently, it is preferable to construct a representation that highlights local transition structure rather than spreading its mass across distant regions of the state space.

We propose the Hierarchical Successor Representation (HSR) to address these limitations. The central idea is to extend the classical SR formulation by incorporating temporal abstractions – formalised through the "options" framework (Sutton et al., 1999). Options define interpretable, temporally extended courses of actions, with distinct policies, initiation and terminal conditions. By reasoning at this higher level of temporal abstraction, HSR aggregates dynamics over interpretable behavioural segments rather than single-step actions, yielding a predictive representation

that is less sensitive to task-induced variations in low-level control. Intuitively, whilst optimal primitive actions may vary across tasks, the high-level strategy (e.g., go through bottleneck states into the next room) remains stable. The HSR leverages this stability to build a task-agnostic representation, enabling more efficient generalisation without requiring expensive re-learning of the state feature maps.

To further promote interpretability and scalability, we seek a compact low-rank basis of these predictive representations. While existing spectral approaches use eigenvectors of transition structure to construct geometry-aware bases (Mahadevan & Maggioni, 2007; Stachenfeld et al., 2017), the resultant components are typically globally supported and sign-indefinite. We therefore employ non-negative matrix factorisation to obtain sparse, localised, and interpretable factors (Lee & Seung, 1999). In multi-compartmental domains, we show that these bases align naturally with topological features such as bottleneck states, and yield a policy-agnostic multi-scale map that can be reused across tasks without sacrificing optimality. These features hence facilitate sample-efficient transfer across reward configurations. Beyond generalisation, we find that HSR can be used to guide intrinsically motivated exploration in sparse-reward environments (Schmidhuber, 1991; Pathak et al., 2017), and enables efficient scaling to larger, procedurally generated maze environments where existing exploration strategies become prohibitively expensive. Importantly, the utility of the proposed HSR framework remains robust with respect to environmental geometry, option definition, and the existence of pre-constructed option set. Collectively, our results suggest that integrating temporal abstraction into predictive representations leads to a robust, interpretable state representation, and provides a practical bridge between model-free efficiency and model-based flexibility (Gershman, 2018): HSR retains the computational advantages of SR-style predictive representations whilst producing hierarchical, interpretable structure that supports robust transfer.

## 2. Preliminaries

**Reinforcement Learning.** We consider the classical RL setup in discrete Markov Decision Processes (MDPs; Sutton & Barto 1998), defined by the tuple, $\langle \mathcal{S}, \mathcal{A}, \mathcal{P}, \mathcal{R}, \gamma \rangle$, where $\mathcal{S}$ and $\mathcal{A}$ denote the state (of size $N$) and action spaces (of size $M$), respectively, $\mathcal{P} : \mathcal{S} \times \mathcal{A} \to \Delta(\mathcal{S})$ is the state transition probability ($\Delta(\mathcal{S})$ denotes a probability measure over $\mathcal{S}$), $\mathcal{R} : \mathcal{S} \to \mathbb{R}$ is the reward function, and $\gamma \in [0, 1]$ is the temporal discounting factor. The goal of an RL algorithm is to learn a policy, $\pi : \mathcal{S} \times \mathcal{A} \to [0, 1]$, such that the agent achieves maximum cumulative future reward starting from any state. Formally, the policy-dependent value and action-value functions are defined as the expected discounted future return, $V^\pi(s) = \mathbb{E}_\pi \left[ \sum_{t=0}^{\infty} \gamma^t \mathcal{R}(s_t) | s_0 = s \right]$.

**Successor Representation.** The SR framework enables

decomposition of the (action) value function into a linear combination of time-collapsed predictive representation and state-dependent reward function (Dayan, 1993).

$$V^\pi(s) = \sum_{s' \in \mathcal{S}} \mathbb{E}_\pi \left[ \sum_{t=0}^{\infty} \gamma^t \mathbb{1}(s_t, s') | s_0 = s \right] \mathcal{R}(s') = \mathbf{M}^\pi \cdot \mathbf{R} , \tag{1}$$

It is clear from the above decomposition that the SR formulation enables rapid generalisation across varying reward configurations, given efficient one-step linear update for policy re-evaluation under new reward functions. However, it is often necessary to simultaneously adapt the policy for achieving optimal performance under new tasks. Drawing upon similar recursive definitions of value functions and the SR, it is hence possible to learn the SR matrix online with temporal-difference (TD) learning (Dayan, 1993). Specifically, given the state-transition tuple $(s_t, a_t, s_{t+1})$, and learning rate $\alpha \in \mathbb{R}$, the update is as following.

$$\hat{\mathbf{M}}_{s_t s'} \leftarrow \hat{\mathbf{M}}_{s_t s'} + \alpha \left( \mathbb{1}(s_t, s') + \gamma \hat{\mathbf{M}}_{s_{t+1} s'} - \hat{\mathbf{M}}_{s_t s'} \right) , \tag{2}$$

**Hierarchical Reinforcement Learning.** Under the hierarchical RL setup, the "option" framework enables the incorporation of temporal abstraction in RL: instead of interacting with the environment at every time step, the agent is able to engage in temporally extended, interpretable action sequences (Sutton et al., 1999). Formally, an option $\omega_i \in \Omega$ is defined by the tuple, $\langle \mathcal{I}_i, \pi_i, \beta_i \rangle$, where $\mathcal{I}_i \subseteq \mathcal{S}$ denotes the initiation set such that an option $\omega_i$ is available in state $s$ if and only if $s \in \mathcal{I}_i$, $\pi_i : \mathcal{S} \times \mathcal{A} \to [0, 1]$ denotes the option-specific policy, and $\beta_i : \mathcal{S} \to [0, 1]$ denotes the option's termination function. Under the option framework, the agent learns a high-level policy, $\mu : \mathcal{S} \times \bar{\mathcal{A}} \to [0, 1]$, where $\bar{\mathcal{A}} = \mathcal{A} \cup \Omega$ denotes the extended action space. Options can be either explicitly pre-defined given expert knowledge (Sutton et al., 1999; Dietterich, 2000; Wang et al., 2021) or automatically discovered (McGovern & Barto, 2001; Bacon et al., 2017; Machado et al., 2017; Jinnai et al., 2019). Here we focus on the "eigenoption" framework (Machado et al., 2017), a task-agnostic option-discovery approach that exploits the transition structure of the environment. Specifically, given the random-walk SR matrix, $\mathbf{M}_0 \in [0, 1]^{N \times N}$, each eigenvector $\mathbf{v}_i \in \mathbb{R}^N$, such that $\mathbf{M}_0 \mathbf{v}_i = \lambda_i \mathbf{v}_i$, defines an option $\omega_i$. The option-specific policy is obtained by solving the MDP with pseudo-reward, defined as following.

$$r_i(s, s') = \mathbf{v}_{i, s'} - \mathbf{v}_{i, s} , \forall s, s' \text{ where } \exists a \text{ s.t. } \mathcal{P}(s'|s, a) > 0 , \tag{3}$$

Upon learning converges with respect to the pseudo-reward function, the optimal pseudo value function, $q_i^*(s, a)$, for all $s$ and $a$, defines the option-specific policy: $\pi_i(a|s) = \mathbb{1}(a, \arg\max_{a'} q_i^*(s, a'))$. The termination function is a binary function such that $\beta_i(s) = 1$ if $q_i^*(s, a') \leq 0 \, \forall a' \in \mathcal{A}$, and 0 otherwise (i.e., local maximum of the pseudo-reward

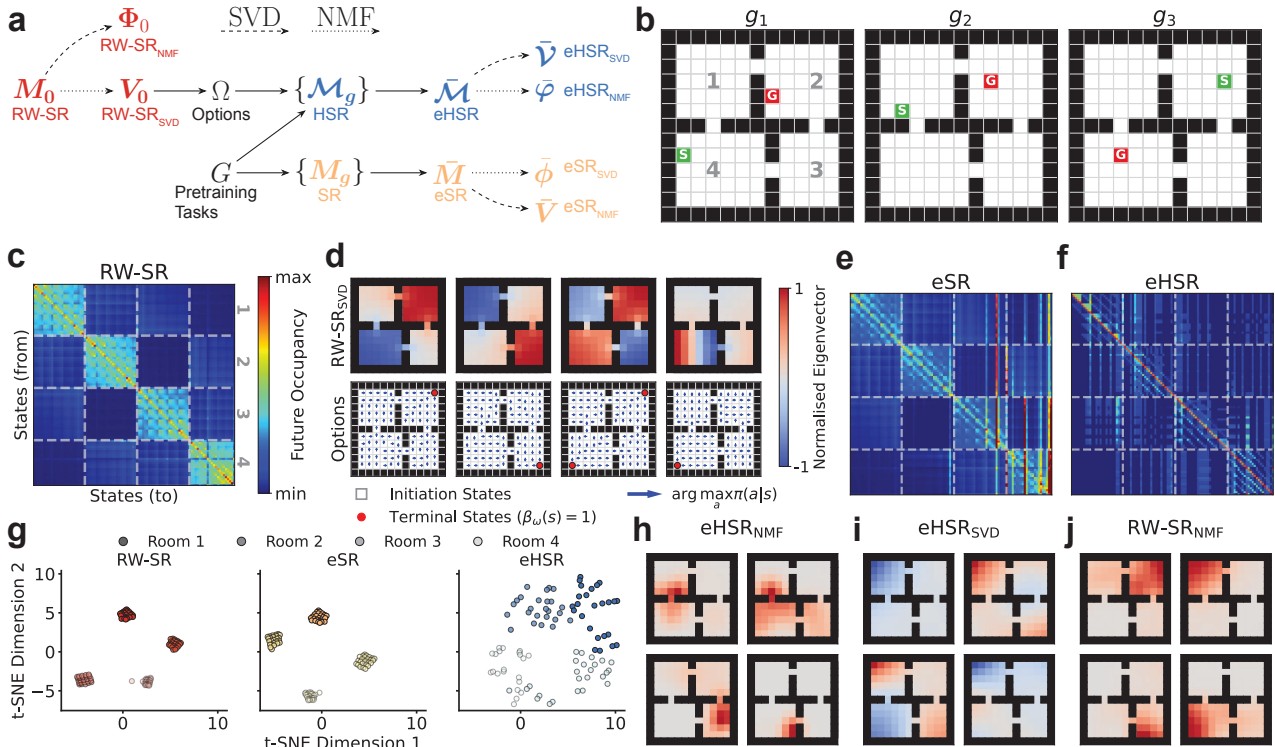

Figure 1. **Temporal abstraction yields hierarchical successor representation. a.** Schematic of the computational process underlying the construction of hierarchical successor representations, and corresponding low-dimensional basis through NMF($\Phi$) and singular value decomposition (SVD; **V**). **b.** Exemplar pretraining regimes ($G$). **c.** SR matrix corresponding to the random-walk policy (RW-SR). Note that state indices were permuted to respect the topological structure of the four-room environment (indicated by gray dashed lines). **d.** Principal eigenvectors (ranked by eigenvalues) of the RW-SR matrix (top), as well as their corresponding eigenoption-specific policies (bottom). **e.** Expected SR matrix (eSR; $\bar{M}$). **f.** Expected HSR matrix (eHSR; $\bar{\mathcal{M}}$). **g.** Two-dimensional t-SNE projections (Maaten & Hinton, 2008) of the row-space of RW-SR (left), eSR (middle), and eHSR (right). **h.** Principal NMF basis vectors (ranked by basis norm) of the eHSR matrix. **i.** Principal eigenvectors of the expected eSR matrix. **j.** Same as **j**, but for the RW-SR matrix. Note that all presented basis vectors were normalised by their corresponding maximum absolute values, hence leaving their signs invariant.

function). The initiation set is defined as the complement to the set of terminal states, $\mathcal{I}_i = \mathcal{S} \backslash \{s|\beta_i(s) = 1\}$. In practice, options are defined by the set of principal eigenvectors of the random-walk SR matrix, which capture the global smoothness of the environment (Shi & Malik, 2000), with decreasing timescales of diffusion (with eigenvalues) in the classical four-room environment (Figure 1d).

## 3. Hierarchical Successor Representation

Despite the SR framework provides a predictive state representation that enables rapid transfer in response to reward changes, reaching optimal transferability still necessitates updating the SR corresponding to policy changes in a model-free fashion (Equation 2). To achieve robust generalisation, we seek a state representation that could retain the predictive nature of the classical SR, but of weaker policy-dependence. We extend the SR formulation via incorporating temporally extended options, leading to the Hierarchical SR (HSR), defined as the predictive state occupancy distribution with respect to the high-level policy, $\mu$ (Equation 1).

$$\mathcal{M}^\mu_{ss'} \equiv \mathbb{E}_\mu \left[ \mathbf{M}^{\bar{a}}_{ss'} + \mathbb{E}_{\bar{a}} \left[ \gamma^{\tau_{s\bar{a}}} \mathcal{M}^\mu_{s_{\tau_{s\bar{a}}} s'} \right] \right], \quad (4)$$

where $\tau_{s\bar{a}} = \mathbb{E}_{\mathcal{P}, \pi_{\bar{a}}} \left[ \sum_{t=0}^\infty \mathbb{1}(\beta_{\bar{a}}(s_t), 0)|s_0 = s \right]$ denotes the expected duration following action $\bar{a}$, and $\mathbf{M}^{\bar{a}}$ denotes the action-specific SR matrix, corresponding to the policy $\pi_{\bar{a}}$, which is defined to be the single-action policy if $\bar{a} \in \mathcal{A}$, and option-specific policy if $\bar{a} \in \Omega$. We have overloaded the notation such that each primitive action is treated as a one-step pseudo-option, such that given any $(s, a) \in \mathcal{S} \times \mathcal{A}$, we have $\mathcal{I}_{sa} = \{s\}$, $\pi_{sa}(a'|s) = \mathbb{1}(a, a')$, and $\beta_{sa}(s') = 1$ if and only if $\mathcal{P}(s'|s, a) > 0$ and 0 otherwise. The recursive definition yields the following HSR Bellman operator, $\mathcal{T}$ (full derivations can be found in A.1).

$$(\mathcal{T}^\mu \mathcal{M})_{ss'} = \mathcal{B}^\mu_{ss'} + \sum_{\tilde{s} \in \mathcal{S}} \mathcal{G}^\mu_{s\tilde{s}} \mathcal{M}_{\tilde{s}s'}, \text{ where}$$

$$\mathcal{B}^\mu_{ss'} = \sum_{\bar{a} \in \bar{\mathcal{A}}} \mu(\bar{a}|s) \mathbf{M}^{\bar{a}}_{ss'}, \quad \mathcal{G}^\mu_{ss'} = \sum_{\bar{a} \in \bar{\mathcal{A}}} \mu(\bar{a}|s) \mathbf{F}^{\bar{a}}_{ss'}, \quad (5)$$

where $\mathbf{F}^{\bar{a}}_{s_j s_j} = \mathbb{E} \left[ \gamma^{\tau_{s_i \bar{a}}} \mathbb{1}(s_{\tau_{s_i \bar{a}}}, s_j)|s_0 = s_i \right]$ is the discounted termination kernel for (pseudo) option $\bar{a}$. With a bit of algebra (Appendix A.1), we can show that $\mathbf{F}^{\bar{a}}$ can be computed analytically as following.

$$\mathbf{F}^{\bar{a}} = \mathbf{M}^{\bar{a}} \text{diag}(\beta_{\bar{a}}), \quad (6)$$

where $\text{diag}(\boldsymbol{\beta}_{\bar{a}})$ is the diagonal matrix with the $i$-th diagonal element equals $\beta_{\bar{a}}(s_i)$. Similar to the standard Bellman operator, $\mathcal{T}$ is a contraction mapping.

**Theorem 3.1** (Contraction of HSR Bellman Operator). *Let $\mathcal{T}$ be the HSR Bellman operator defined by Equation 5. For any discount factor $\gamma < 1$ and option durations $\tau \geq 1$, $\mathcal{T}$ is a contraction mapping with respect to the max-norm (proof can be found in A.2).*

$$||\mathcal{T}^\mu \mathcal{M} - \mathcal{T}^\mu \mathcal{M}'||_\infty \leq \gamma ||\mathcal{M} - \mathcal{M}'||_\infty \qquad (7)$$

*for any $\mathcal{M}$ and $\mathcal{M}'$.*

Consequently, the HSR Bellman operator defines an equivalent iterative update rule for learning the HSR online, similar to the TD-learning for standard SR (Equation 2): $\hat{\mathcal{M}}_{s_t s'} \leftarrow \hat{\mathcal{M}}_{s_t s'} + \alpha \delta_t^{\mathcal{M}}$. Given the state-transitions trajectory following the (pseudo) option, $\bar{a}_t$, $(\{(s_{t+k}, \bar{a}_{t+k})\}_{k=0}^{\tau_t}, s_{t+\tau_t+1})$[1], the TD-error defined as following.

$$\delta_t^{\mathcal{M}} = \left( \sum_{k=0}^{\tau_t} \gamma^k \mathbb{1}(s_{t+k}, s') + \gamma^{\tau_t+1} \hat{\mathcal{M}}_{s_{\tau_t+1} s'} - \hat{\mathcal{M}}_{s_t s'} \right) \qquad (8)$$

While the HSR inherently supports online TD learning to track a changing policy, this adaptation can be sensitive to transient behavioural shift. Therefore, in transfer scenarios where pre-exposure to the environment is possible via a distribution of pre-training tasks (Figure 1b), we advocate for a stable offline construction. Specifically, we compute the *Expected* HSR (eHSR) by averaging the HSRs derived from optimal policies in these pre-training tasks, yielding a generalised basis that further relaxes the policy dependence in the constructed predictive representation.

**Low-Rank Basis of HSR.** Constructing low-dimensional state representations via the decomposition of transition dynamics has long served as an effective representation learning approach in RL. Seminal works, such as Proto-value functions (PVFs; Mahadevan & Maggioni 2007) and spectral successor features (Stachenfeld et al., 2017), rely on eigenvectors of the Laplacian operator and SR, respectively, to define a geometric basis. Following this paradigm, we also consider using eigenvectors of the HSR as state representations, which indeed captures the globally supported smooth dynamics (Figure 1i). However, the HSR introduces a temporally extended, topology-aware predictive representation, which is fundamentally different from the classical SR. Classical SRs exhibit topological collapse: the predictive features of intra-compartment states are tightly clustered, reducing the effective dimension within each topological region to a single point mass (Figure 1g). This implies the classical SR is numerically low-rank, a regime where singular value decomposition (SVD) is optimal for

capturing the global modes of variation (also indicated by the eigenvalue distribution; Figure S5a).

In contrast, HSR dynamics is piecewise-smooth, characterised by manifolds with effective intra-compartment dispersion, whilst preserving the inter-compartment differences (Figure 1g). Whilst SVD can compress the HSR, its orthogonal basis vectors inevitably "smear" the multi-scale structure of these manifolds across global components, leading to oscillatory ringing artefacts induced by orthogonal low-rank reconstruction (also known as the Gibbs phenomenon; Figure 1i; Stephane 1999) and heavier-tailed spectral components (Figure S5a). Consequently, we employ non-negative matrix factorisation (NMF) to extract the $K$-dimensional sparse, localised basis (Lee & Seung, 1999; Hoyer, 2004).

$$\mathcal{M} = \boldsymbol{\varphi} \cdot \mathbf{H}, \text{ where } \boldsymbol{\varphi} \in \mathbb{R}^{N \times K}, \mathbf{H} \in \mathbb{R}^{K \times N} \text{ s.t. } \boldsymbol{\varphi}, \mathbf{H} \geq 0, \qquad (9)$$

The locally dispersed geometry of the HSR enables the NMF to discover the ground-truth underlying generative parts (Figure 1h; Donoho & Stodden 2003). However, the feature collapse in standard SRs makes "parts-based" decomposition via NMF ill-posed, due to the fact that there is insufficient intra-compartment variance to define distinct basis components, leading to trivial and overlapping factorisations (Figure 1j; Figure S1b). Schematics of the full computational process for deriving HSR and its low-rank decomposition can be found in Figure 1a, and corresponding pseudocode can be found in Algorithm S1.

## 4. Results

We evaluate the efficacy of the proposed HSR-based state representations in facilitating robust transfer, interpretable state abstractions, and scalable exploration. The empirical experiments aim to test three central hypotheses: (a) HSR features are more robust to task-induced policy changes than standard SRs under transfer-learning scenarios; (b) the topological structure of HSR is uniquely amenable to NMF decomposition, yielding interpretable state features that facilitates stronger generalisation than existing spectral-based decomposition methods; and (c) these temporally extended predictive features enable efficient and scalable exploration in large, procedurally generated maze environments. All agents in transfer experiments were implemented based on Q-learning with linear function approximation. Unless otherwise stated, empirical evaluations are based on 20 random seeds. Implementation details can be found in Appendix C.

### 4.1. HSR facilitates robust few-shot transfer.

We first investigate the robustness of HSR under changing reward configurations in the classical four-room environment (Figure 2a). Here we assume both SRs and HSRs are learned online as policy changes, both within and between tasks. To demonstrate the utility of canonical HSR features,

---

[1]Note that we have overloaded notations to denote $\tau_t := \tau_{s_t \bar{a}_t}$.

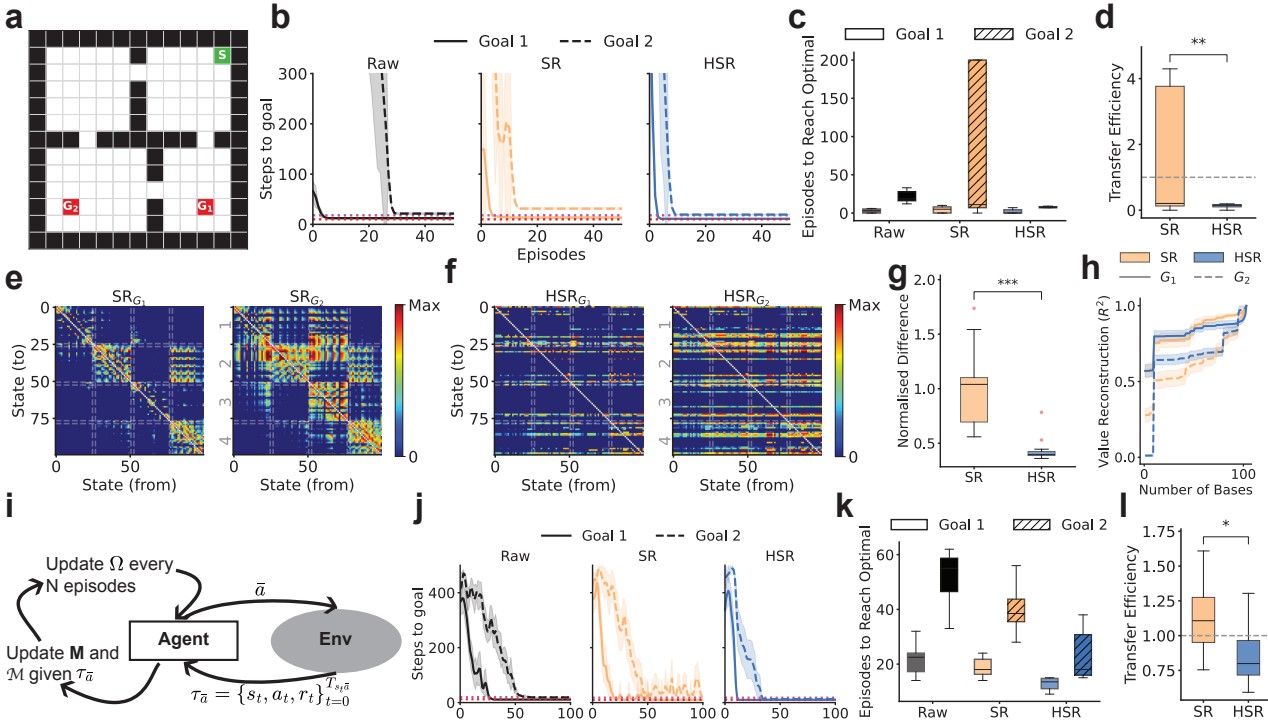

*Figure 2.* **HSR provides a stable state representation and enables sample-efficient transfer across tasks with shared transition dynamics. a.** Schematics of the four-room environment. All agents were firstly trained to reach $G_1$, and subsequently transferred to the new task with goal location $G_2$, both from a fixed start location. **b.** Number of steps to reach the goal location (mean $\pm$ s.e.) for Q-learning agents with linear function approximation, given different state representations. All state representations (apart from one-hot representation) were simultaneously trained with the value function. Dashed horizontal red and magenta lines indicate optimal number of steps to reach $G_1$ (10) and $G_2$ (18) from the shared start state. **c.** Number of training episodes to reach optimal performance in the $G_1$ and $G_2$ tasks, for all agents in **b** (number set of 200 if agents fail to reach optimal performance within 200 episodes). **d.** Transfer efficiency (normalised ratio between number of training episodes to reach optimal performance in $G_1$- and $G_2$-tasks) for SR- and HSR-based agents (two-sided two-sample t-test; $p = 0.008$, $df = 38$). **e.** SR matrices (log-scale for visual clarity) after the corresponding agent were trained to reach optimal performance in $G_1$ (left) and $G_2$ (right) tasks. Note that rows and columns of SR matrices are permuted to restore the local topological structure of the environment (Figure 1b). We omit showing diagonal elements for visual clarity. **f.** Same as **e**, but for HSR matrices. **g.** Degrees of change in predictive representation $\left(\frac{||M_1 - M_2||_F^2}{||M_1||_F^2}\right)$ after agents were trained in $G_1$ and $G_2$ tasks, for SR and HSR matrices, respectively (two-sided two-sample t-test; $p < 0.001$, df = 38). **h.** Reconstruction $R^2$ scores of ground-truth optimal value functions (computed via dynamic programming) for $G_1$ and $G_2$ tasks given varying number of SR/HSR basis after agents are trained to reach optimal performance in $G_1$ tasks. **i.** Schematics of the HSR framework with online-constructed option set. **j.** Same as **b**, but for agents with online-constructed option sets. **k.** Same as **c**, but for agents in **j**. **l.** Same as **d**, but for SR and HSR agents with online constructed options (two-sided two-sample t-test; $p = 0.026$, $df = 38$.).

we take rows of SR and HSR matrices as state features. We also implement an agent with fixed one-hot state encoding ("Raw") as the zero-transfer baseline. For fair comparison with the HSR-based agents, all agents were trained to choose actions from the augmented action space, consists of primitive actions and the set of 8 principal eigenoptions (Figure S1a; note that standard SR matrices are still updated on a stepwise basis during the course of executing options). Agents were trained to navigate to a specific terminal goal state ($G_1$) until convergence, and subsequently transferred to navigate to a new goal ($G_2$) starting from the same initial location[2]. Transitions into goal states yield a reward of 1,

whereas all other transitions yield 0 reward.

Both the standard and hierarchical SR features enable more sample-efficient transfer relative to the one-hot state encoding baseline (Figure 2b, c). Consistent with the policy-dependence limitation of standard SRs, that the policy re-optimisation invalidates the established predictive representations, we observe that agents utilising row features of random-walk SR (RW-SR$_{row}$) suffer from significant performance degradation upon goal-switching (Figure 2b, c). In contrast, agents utilising the HSR row features exhibit strong few-shot transfer, rapidly adapting to the new goal with significantly fewer episodes upon transfer. Consistent with the visual examination of learning curves and our intuitions, HSR agents achieve a significantly higher transfer

---

[2]Goal locations are selected such that they do not coincide with any of the terminal states of available eigenoptions.

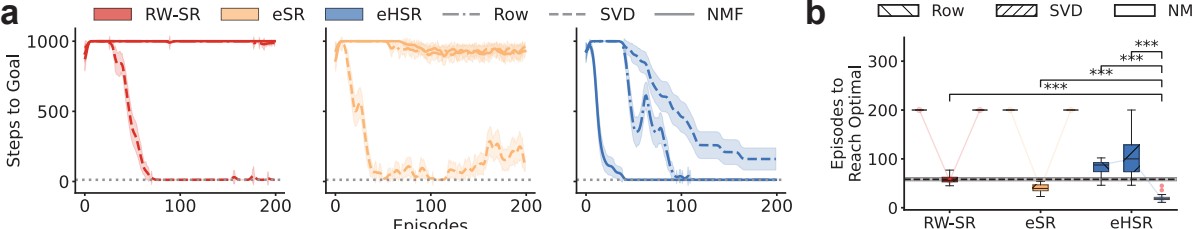

*Figure 3.* **NMF basis of HSR supports sample-efficient transfer. a.** Training curves (left) and number of training episodes to reach optimal performance (right) in $G_1$ tasks (Figure 2a) for Q-learning agents with linear function approximation, given different low-dimensional basis as state representations. All agents were assumed to have received necessary pretraining for constructing base matrices (SR/HSR) before corresponding low-dimensional basis were extracted. Gray dotted line indicates optimal number of steps to reach $G_1$. **b.** Number of training episodes required to reach optimal performance for all agents in **a** (two-sided Wilcoxon signed-rank test; RW-SR$_{SVD}$ vs HSR$_{NMF}$: $p = 7.91 \times 10^{-17}$; ESR$_{SVD}$ vs HSR$_{NMF}$: $p = 2.93 \times 10^{-9}$; HSR$_{SVD}$ vs HSR$_{NMF}$: $p = 6.73 \times 10^{-10}$; HSR$_{NMF}$ vs HSR$_{Row}$: $p = 4.59 \times 10^{-18}$; $N = 20$ for all tests). Gray dashed horizontal line indicates the same number for the baseline agent with one-hot state encoding (shaded area indicates s.e.).

efficiency (Figure 2d), where transfer efficiency is quantified by the ratio of number of training episodes to reach optimal performance in $G_1$ and $G_2$ tasks, normalised by the same numbers given the baseline agent ($\frac{N_{G_2}^{HSR/SR}/N_{G_2}^{Raw}}{N_{G_1}^{HSR/SR}/N_{G_1}^{Raw}}$)..

We attribute this robustness to the stability of the underlying representation, arising from the temporal extended components in constructing the HSR that decouples predictive representations from transient variations in low-level control. Comparing the predictive representation after learning asymptotes in the two tasks reveals that standard SR matrices undergo drastic reorganisation in order to conform with the new optimal policy (Figure 2e), whereas the HSR matrices are less variable following policy changes (Figure 2f). Quantitative comparison of magnitudes of change confirmed that the relative change in HSR matrices following policy adaptation was significantly lower than that of the SR matrix (Figure 2g). Such stability translates to elevated capacity in value estimation. We compare the reconstruction of the optimal value functions of both the current task ($G_1$) and the unseen task ($G_2$) give established state features after learning in the $G_1$ task, with varying numbers of bases. We observe that SR and HSR explain the optimal value function of the current task equally well, but the HSR basis consistently yields significantly more accurate reconstruction of optimal value functions in unseen tasks (Figure 2h). This suggests that the HSR features span a subspace that is more robustly aligned with the manifold of potential value functions across the task distribution. Note that the improved transferability and stability of HSR features, relative to standard SR features, are not artefacts of the online learning process. This is further corroborated by similar empirical results when agents are trained with pre-trained state representations (Figure S2). Moreover, the transfer efficiency induced by HSR features remains robust across environments with different geometry (Figure S3) and option classes beyond eigenoptions (Figure S4).

**Online Option Construction.** Notably, the HSR frame-

work is not contingent on the access to a set of pre-existing options, and can incorporate online-constructed option set (Figure 2i; see Appendix B.2 and Algorithm S2 for further details) whilst retaining its utility in supporting stronger transfer efficiency (Figure 2j-l).

### 4.2. HSR-NMF basis facilitates robust transfer.

We next evaluate the robustness of low-dimensional features derived from the predictive representations. To ensure fair comparison, we include an additional baseline which constructs an "expected SR" (eSR) given the same set of pre-training tasks used for constructing the expected HSR (Figure 1a, bottom branch; see also Appendix C.1 and Algorithm S3). We hypothesised that while SVD is optimal for the smooth and diffusive dynamics of standard SRs, the piecewise-smooth topology of HSR is better captured by NMF decomposition.

Conforming with our hypothesis, for RW-SR and eSR, SVD-based features enables efficient transfer (given pre-training), whereas NMF fails to produce a useful basis, resulting in poor performance (Figure 3a). We attribute the failure of NMF on SR features to "feature collapse" given the lack of intra-compartment variability (Figure 1g). Crucially, the pattern is reversed for HSR: HSR-NMF features not only proved superior to HSR-SVD, but also outperformed all SR-based baselines, in terms of sample efficiency of transfer (Figure 3a, b). Notably, row features of HSR yield more rapid learning than corresponding SVD features, whereas those of standard SRs performed poorly.

We hypothesised that the performance advantage of HSR-NMF stems from the sparsity and stability of these bases. Indeed, through examining the temporal trace of basis activation along a circular trajectory around the environment (Figure 4a), we confirm that while spectral decomposition and standard SR features exhibit entangled, noisy activations that fluctuate rapidly, HSR-NMF reveals a sparse code that uniformly tiles the state space (Figure 4b). The localised

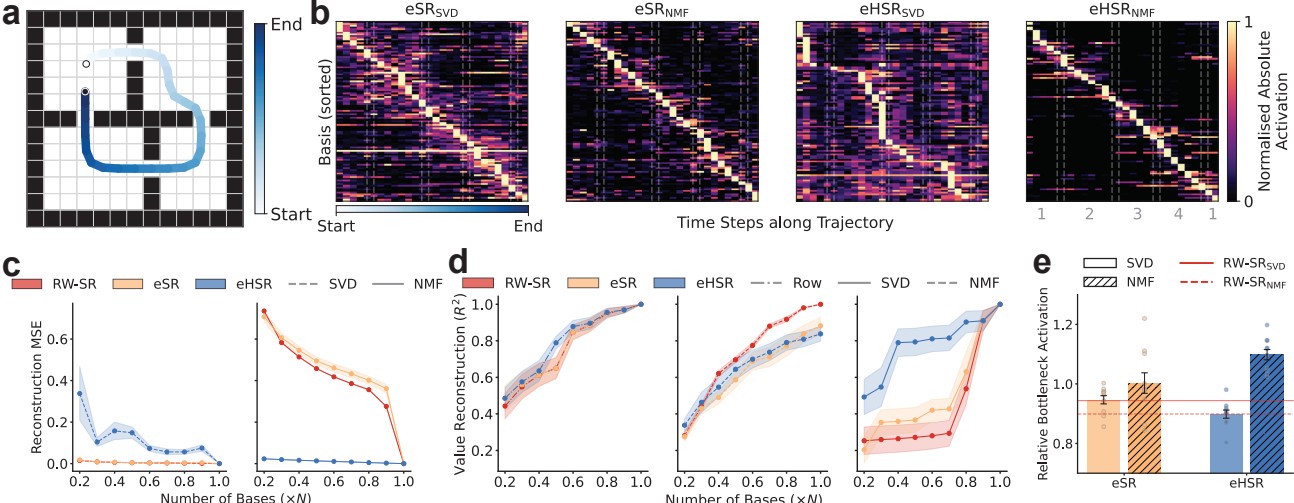

*Figure 4.* **HSR-NMF basis yield a sparse, robust, and interpretable state representation. a.** Example trajectory in the four-room environment. **b.** Activation (normalised) of all basis at each timestep along the example trajectory for eSR$_{SVD}$, eSR$_{NMF}$, HSR$_{SVD}$, HSR$_{NMF}$ (from left to right). Gray numbers below the rightmost panel indicates which room the corresponding trajectory segment is in. **c.** Reconstruction mean-squared error of predictive representations given corresponding low-dimensional features, as functions of varying number of bases. **d.** Reconstruction $R^2$ score of optimal value functions with respect to randomly selected goal locations, given different state features with varying basis size. **e.** Relative bottleneck activation (mean activation at bottleneck states / mean activation at non-bottleneck states) of low-dimensional features of SR and HSR.

features support efficient approximation of arbitrary value function over the state space (Figure 4c), whilst retaining the dynamics-aware predictive nature. In contrast, the globally supported, oscillatory bases given spectral decomposition yield value learning difficult, as increasing value estimates in one room likely cause simultaneous decrease in estimated values in another room, a classical "whack-a-mole" issue. The stability of the basis is underpinned by the intrinsic compressibility of the HSR matrix. Crucially, we observe that HSR-NMF matches the reconstruction efficiency of SR-SVD (Figure 4d), despite its heavy-tailed spectral distribution (Figure S5a). Hence, despite the additional positiveness penalty, the piecewise smooth structure of the HSR enables NMF to discover faithful low-dimensional representation whilst achieving SVD-level compression efficiency. The discovered latents additionally exhibit stronger interpretability, with elevated activations at bottleneck states (Figure 4e), the key states governing successful navigation underlying arbitrary reward configurations in the state space.

### 4.3. HSR and Generalised Policy Improvement

An related class of methods for addressing the transfer efficiency of RL agents through the lens of representation learning are the Generalised Policy Improvement (GPI) with SR (Figure 5a; Barreto et al. 2017; 2018; Carvalho et al. 2023). SR-GPI algorithms leverage a similar two-stage process, solve a set of pre-training tasks (Figure 1b) and maintains a pool of pre-trained policies, each yielding policy-specific SR matrix, $\mathbf{M}^{\pi_g}$. In the downstream transfer task with state-dependent reward vector $\mathbf{r}_*$, each pretrained policy is re-evaluated given corresponding SRs

(Equation 1), and action selection is then performed via GPI: $a_*(s) = \arg\max_a \left[ \max_g q_*^{\pi_g}(s, a) \right]$. Importantly, SR-GPI and HSR tackle transfer learning from orthogonal yet complementary angles, focusing on algorithmic and representational advancements, respectively.

We empirically evaluate the transfer efficiency of SR-GPI, augmented with linear function approximation given different state representations (Figure 5b,c). We note that SR-GPI requires access to the ground-truth reward function. To ensure fair comparison, $\mathbf{r}_*$ is not provided *a priori*, but needs to be learned online upon first encounter of the single goal location (see Appendix C.2 for further details). Under this setup, the canonical SR-GPI with one-hot state representation performed comparably to the HSR agents that do not perform policy composition. Moreover, combining SR-GPI with HSR state representations, either row or NMF features, yields stronger transfer efficiency, hence supporting our argument that HSR and GPI offer complementary benefits: policy-composition on the one hand, and generalisable representation on the other.

### 4.4. Scalable exploration with HSR.

We finally investigate whether HSRs facilitate stronger RL beyond providing a robust state representation. Recent works demonstrate the utility of standard SRs in constructing effective intrinsic motivation for exploration in sparse-reward environments (Machado et al., 2020; Yu et al., 2023). We specifically consider two SR-based intrinsic rewards: row-norms as a proxy for state visitation count (Machado et al., 2020), and the successor-predecessor intrinsic explo-

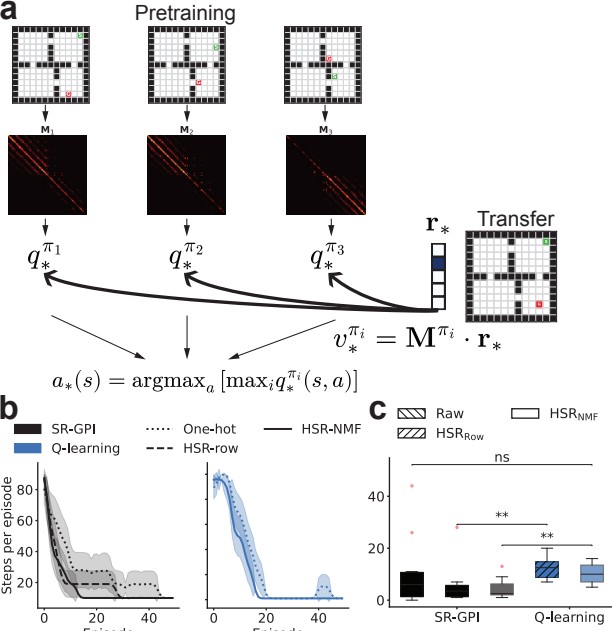

$$a_*(s) = \operatorname{argmax}_a \left[ \max_i q_*^{\pi_i}(s, a) \right]$$

*Figure 5.* **HSR-NMF yields comparable transfer efficiency to SR-based Generalised Policy Improvement (GPI) algorithm. a.** Schematic of SR-GPI algorithms (Barreto et al., 2017). **b.** Number of steps to reach the goal over the first 50 training episodes (mean $\pm$ s.e.; 10 random seeds) of SR-GPI agent and linear Q-learning under one-hot and different HSR-based state representations. **c.** Number of training episodes required to reach optimal performance for all agents in **b** (two-sided two-sample paired t-test; SR-GPI with one-hot state representations vs linear Q-learning with HSR$_{\text{NMF}}$ features: $p = 0.199$; SR-GPI with HSR$_{\text{Row}}$ features vs linear Q-learning with HSR$_{\text{Row}}$ features: $p = 0.003$; SR-GPI with HSR$_{\text{NMF}}$ features vs linear Q-learning with HSR$_{\text{NMF}}$ features: $p = 0.007$; $df = 18$).

ration (SPIE) that combines both temporally prospective and retrospective information (Yu et al., 2023).

$$r_{\text{SR}}(s) = \frac{1}{||\mathbf{M}_{s:}||_1}, \quad r_{\text{SPIE}}(s) = \mathbf{M}_{ss'}^2 - ||\mathbf{M}_{:s'}||^2, \quad (10)$$

We hypothesise that the temporally extended structure of HSR could support more efficient exploration comparing to standard SRs. We hence implement SARSA-based agents with these intrinsic rewards, computed given either the standard or hierarchical SR. We test agents in procedurally generated random mazes of varying sizes (Figure 6a).

We firstly evaluate the exploration efficiency of implemented agents – quantified through the rate at which state coverage increases with experience – in pure exploration tasks (in the complete absence of extrinsic reward). Building upon the established superiority of SPIE objectives over SR-norm objectives (Yu et al., 2023), we observe that HSR-augmented objectives significantly outperform their SR-based counterparts, covering a larger fraction of the state space within a fixed budget (Figure 6b, left). The gap in exploration efficiency becomes increasingly pronounced as the environment size grows. Whilst the asymptotic coverage of SR-SPIE de-

graded drastically in larger mazes, HSR-SPIE maintained high state coverage (Figure 6c). In sparse-reward navigation tasks (only non-zero reward for transitions into randomly selected goal state), HSR-augmented intrinsic objectives achieved high probability of discovering the goal location (Figure 6b, right). Somewhat surprisingly, despite HSR-norm intrinsic reward yields slow state coverage, it is nevertheless more effective than both SR-based objectives in sparse-reward navigation tasks. These results suggest that by incorporating extended temporal horizon into predictive representations, the HSR enables agents to "escape" from local diffusive barriers, facilitating extended exploration in environments with complex topology where single-step predictive methods become inefficient.

Finally, consistent with the observation that the HSR yields sustained improvement in transfer efficiency in the absence of pre-constructed option set, we confirmed empirically that intrinsic rewards facilitated by HSRs learned given online-constructed options supports more efficient exploration than SR-based intrinsic rewards (Figure S6).

## 5. Related Work

**Generalisation in RL.** Recent efforts to improve the generalisability of RL agents have been predominantly based on algorithmic advances. These approaches largely focus on enforcing robustness via novel regularisation and optimisation objectives (Farebrother et al., 2018; Cobbe et al., 2019; Igl et al., 2019), or by artificially expanding the diversity of training distributions through data mixing (Wang et al., 2020) and augmentation (Laskin et al., 2020a; Yarats et al., 2021). Moreover, an influential line of works based on generalised policy improvement address transfer learning through recycling and composing pre-trained policies (Barreto et al., 2017; 2018; Carvalho et al., 2023). Notably, the performance of GPI-based algorithms critically hinges on the two-stage process involving pre-training on a set of related tasks, whereas the HSR framework is robust with respect to the existence of such pre-training phase (Figure 2).

Alternative to the algorithmic perspective, existing representation learning methods have typically addressed generalisation through the lens of invariance with respect to background noise and local perturbations, leveraging techniques such as bisimulation metrics (Zhang et al., 2020) and contrastive learning (Agarwal et al., 2021; Laskin et al., 2020b). However, true multi-task generalisation requires a task-equivariant representation that captures the underlying structure of the environment, hence facilitating transfer across varying task-induced reward configurations. Our HSR framework bridges this gap by enforcing a temporally abstract, sparse geometry, whilst being robust to policy shifts. Crucially, the parallel advancements in algorithmic and representational methods are complementary, and their combination is expected to result in stronger generalisation.

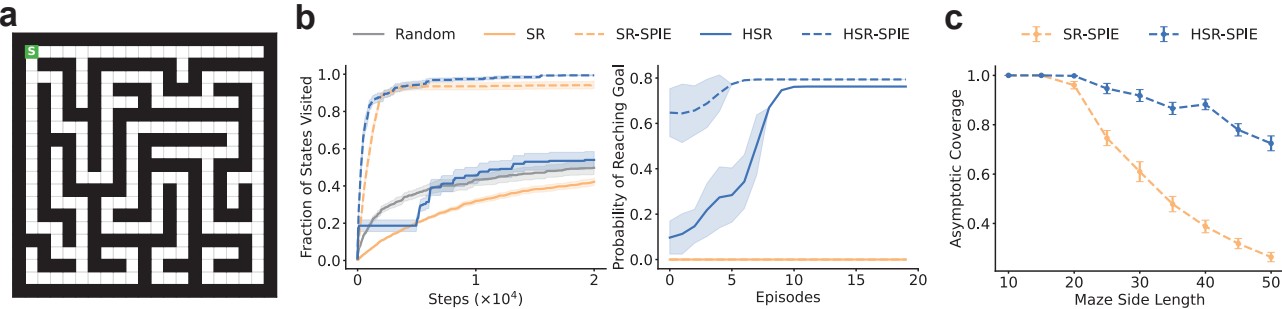

*Figure 6.* **Hierarchical temporal abstraction enables scalable intrinsically motivated exploration. a.** Exemplar procedurally generated random maze environment. **b.** Learning curves (mean ± s.e.) of different agents (see main text) in terms of pure exploration (in the absence of extrinsic reward; left) and goal-directed navigation (with only non-zero reward at randomly selected goal locations; right). **c.** Asymptotic state coverage (after $10^5$ interaction steps; mean ± s.e.) for SR-SPIE and HSR-SPIE agents, as a function of maze size.

**Hierarchical intrinsic exploration.** Solving sparse-reward tasks requires efficient exploration that extends beyond local dithering. While existing intrinsic motivations based on prediction error (Pathak et al., 2017) and state visitation counts (Bellemare et al., 2016; Machado et al., 2020) drive effective local exploration, they often fail to traverse topological bottlenecks and cannot induce sustained exploration in response to dynamic reward structure, especially in larger environments (Yu et al., 2023). In their seminal work, Kulkarni et al. (2016) addressed this by integrating hierarchical RL with intrinsic motivation, demonstrating that agents reasoning at higher-level of temporal abstraction could solve hard-exploration tasks where "flat" agents failed. Our work shares this fundamental insight, such that temporal abstraction is necessary for driving scalable and temporally extended exploration. However, we approach the problem through the lens of representation learning rather than explicit hierarchical control. The multi-scale map is built into the construction of state features, subsequently enables exploration signals to "jump" across local diffusive barriers without requiring explicit subgoal selection.

## 6. Discussion

We introduce hierarchical successor representation, an extension of the classical SR framework to incorporate temporal abstractions. The temporally extended predictive representations yield robust state features with respect to task-induced changes in reward structures and associated policies, hence facilitating stronger transferability. Low-rank decomposition of HSR features based on NMF further supports construction of sparse, interpretable state representations, again leading to improved generalisability. HSR additionally supports efficient and scalable intrinsically motivated exploration, extending its advantages beyond the transfer regime.

We have focused our empirical evaluations on discrete, tabular environments in order to isolate the topological and representational properties of HSR under controlled conditions. Extension to continuous MDPs will inevitably incur deep-learning-based function approximation, and introduce complex confounding factors, such as non-stationary feature learning (Lyle et al., 2019) and parameter interference (Bengio et al., 2013). These confound can obscure the specific representational characteristics we aim to formalise. Having established their structural advantages confirmed under tractable settings in the current work, a natural future direction is to integrate HSR into deep RL systems, where its stable, policy-agnostic properties could serve as a robust objective for representation learning in high-dimensional and continuous domains (Jaderberg et al., 2016; Machado et al., 2017).

Here we have primarily considered settings in which options are acquired through pre-exposure to the environment. Consequently, the strongest transfer results presented here concern tasks with shared transition dynamics. However, HSR also remains effective when no option set is available a priori, using online option construction. The procedure used here is deliberately simple and heuristic; developing principled algorithms for iterative option discovery, and characterising their theoretical consequences, remains an important direction for future work. This will be particularly important in non-stationary environments, where agents must revise their option set as transition structure changes, allowing the state representation to adapt online while preserving the generalisation benefits of hierarchical predictive structure.

So far we have interpreted NMF basis of the HSR primarily through the lens of low-rank decomposition, emphasising data compression. However, a distinct advantage of NMF is its capability to extract an *over-complete* basis, which has promising implications in how the neural system performs efficient coding (Olshausen & Field, 1997; Lewicki & Sejnowski, 2000), particularly when combined with the predictive map theory of hippocampal neuronal firing through the lens of successor representation (Stachenfeld et al., 2017; Yu et al., 2020; Piray & Daw, 2021). Exploring HSR-NMF in the over-complete regime presents a promising avenue for scaling to environments with repetitive or aliased substructures, where spectral bases often fail to discriminate distinct states.

## Acknowledgement

We thank Dominik Straub and anonymous reviewers for helpful discussions and comments. This work was supported by a Wellcome Trust Investigator Award in Science (212262/Z/18/Z; M.L.) and the Simons Collaboration on Ecological Neuroscience (M.L.). Authors declare no conflict of interest.

## Impact Statement

This paper presents work aiming to advance the field of reinforcement learning and representation learning. There are many potential societal consequences of our work, none of which we feel must be specifically highlighted here.

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

# A. Derivations and Proofs

## A.1. Derivation of HSR.

As a reminder, the HSR is defined as the expected discounted future occupancy under some high-level policy, $\mu : \mathcal{S} \times \bar{\mathcal{A}} \to [0, 1]$, where $\bar{\mathcal{A}} = \mathcal{A} \cup \Omega$ denotes the augmented action space.

$$
\begin{aligned}
\mathcal{M}_{ss'}^{\mu} &= \mathbb{E}_{\mu} \left[ \sum_{t=0}^{\infty} \gamma^t \mathbb{1}(s_t, s') | s_0 = s \right] \\
&= \sum_{\bar{a} \in \bar{\mathcal{A}}} \mu(\bar{a}|s) \mathbb{E}_{\bar{a}} \left[ \sum_{t=0}^{\tau_{s\bar{a}}-1} \gamma^t \mathbb{1}(s_t, s') + \gamma^{\tau_{s\bar{a}}} \sum_{k=0}^{\infty} \gamma^k \mathbb{1}(s_{\tau_{s\bar{a}}+k}, s') | s_0 = s, \bar{a} \right] \\
&= \sum_{\bar{a} \in \bar{\mathcal{A}}} \mu(\bar{a}|s) \underbrace{\mathbb{E}_{\bar{a}} \left[ \sum_{t=0}^{\tau_{s\bar{a}}-1} \gamma^t \mathbb{1}(s_t, s') | s_0 = s, \bar{a} \right]}_{\mathbf{M}_{ss'}^{\bar{a}}} + \sum_{\bar{a} \in \bar{\mathcal{A}}} \mu(\bar{a}|s) \sum_{\tilde{s} \in \mathcal{S}} \underbrace{\mathbb{E}_{\bar{a}} \left[ \gamma^{\tau_{s\bar{a}}} \mathbb{1}(s_{\tau_{s\bar{a}}}, \tilde{s}) | s_0 = s, \bar{a} \right]}_{\mathbf{F}_{s\tilde{s}}^{\bar{a}}} \mathcal{M}_{\tilde{s}s'}^{\mu} \\
&= \mathcal{B}_{ss'}^{\mu} + \sum_{\tilde{s} \in \mathcal{S}} \mathcal{G}_{s\tilde{s}}^{\mu} \mathcal{M}_{\tilde{s}s'}^{\mu} ,
\end{aligned}
\tag{S1}
$$

where we conditioned on the termination state $\tilde{s}$ in the second term of the penultimate line, then leverage the definition of the *expected intra-option SR* and *hierarchical continuation kernel* (Equation 5) as following.

$$
\mathcal{B}_{ss'}^{\mu} = \sum_{\bar{a} \in \bar{\mathcal{A}}} \mu(\bar{a}|s) \mathbf{M}_{ss'}^{\bar{a}} , \quad \mathcal{G}_{s\tilde{s}}^{\mu} = \sum_{\bar{a} \in \bar{\mathcal{A}}} \mu(\bar{a}|s) \mathbf{F}_{s\tilde{s}}^{\bar{a}} ,
\tag{S2}
$$

where we define $\mathbf{M}_{ss'}^{\bar{a}} = \mathbb{E}_{\pi_{\bar{a}}} \left[ \sum_{t=0}^{\tau_{s\bar{a}}-1} \gamma^t \mathbb{1}(s_t, s') | s_0 = s \right]$ as the intra-option SR.

We introduce the *discounted termination kernel*, $\mathbf{F}^{\bar{a}}$, defined as following.

$$
\begin{aligned}
\mathbf{F}_{ss'}^{\bar{a}} &= \sum_{t=0}^{\infty} \gamma^t \mathbb{P}(\bar{a} \text{ terminate at } s' \text{ at time } t | s_0 = s) \\
&= \left( \sum_{t=0}^{\infty} \gamma^t \mathbb{P}^{\bar{a}}(s_t = s' | s_0 = s) \right) \cdot \beta_{\bar{a}}(s') \\
&= \left[ \mathbf{M}^{\bar{a}} \cdot \text{diag}(\boldsymbol{\beta}_{\bar{a}}) \right]_{ss'} ,
\end{aligned}
\tag{S3}
$$

Hence, the second term in the recursive definition (Equation S1) can be re-expressed as following.

$$
\mathbb{E}_{\mu, \bar{a}} \left[ \gamma^{\tau_{s\bar{a}}} \mathcal{M}_{s_{\tau_{s\bar{a}}} s'}^{\mu} \right] = \sum_{\tilde{s} \in \mathcal{S}} \mathbb{E}_{\bar{a}} \left[ \gamma^{\tau_{s\bar{a}}} \mathbb{1}(s_{\tau_{s\bar{a}}}, \tilde{s}) | s_0 = s \right] \mathcal{M}_{\tilde{s}s'}^{\mu} = \sum_{\tilde{s} \in \mathcal{S}} \mathbf{F}_{s\tilde{s}}^{\bar{a}} \mathcal{M}_{\tilde{s}s'}^{\mu} ,
\tag{S4}
$$

We hence have derived the Bellman recursion for HSR.

$$
\begin{aligned}
\mathcal{M}_{ss'}^{\mu} &= \underbrace{\sum_{\bar{a} \in \bar{\mathcal{A}}} \mu(\bar{a}|s) \mathbf{M}_{ss'}^{\bar{a}}}_{\mathcal{B}_{ss'}^{\mu}} + \sum_{\tilde{s} \in \mathcal{S}} \underbrace{\left( \sum_{\bar{a} \in \bar{\mathcal{A}}} \mu(\bar{a}|s) \mathbf{F}_{ss''}^{\bar{a}} \right)}_{\mathcal{G}_{ss''}^{\mu}} \mathcal{M}_{s''s'}^{\mu} , \\
\mathcal{T}^{\mu} \mathcal{M} &= \mathcal{B}^{\mu} + \mathcal{G}^{\mu} \mathcal{M} ,
\end{aligned}
\tag{S5}
$$

### A.2. Proof of Theorem 3.1

We here provide the proof that the HSR Bellman operator is a contraction mapping under max-norm.

*Proof.* Consider two arbitrary matrices $\mathbf{M}, \mathbf{M}'$. From Equation 5, the element-wise difference is:

$$|(\mathcal{T}^{\mu}\mathbf{M})_{ss'} - (\mathcal{T}^{\mu}\mathbf{M}')_{ss'}| = \left| \sum_{\tilde{s}\in\mathcal{S}} \mathcal{G}^{\mu}_{s\tilde{s}}(\mathbf{M}_{\tilde{s}s'} - \mathbf{M}'_{\tilde{s}s'}) \right| \leq \sum_{\tilde{s}\in\mathcal{S}} \mathcal{G}^{\mu}_{s\tilde{s}}||\mathbf{M} - \mathbf{M}'||_{\infty} \leq \left( \sup_{s} \sum_{\tilde{s}\in\mathcal{S}} \mathcal{G}^{\mu}_{s\tilde{s}} \right) ||\mathbf{M} - \mathbf{M}'||_{\infty} \quad \text{(S6)}$$

The supremum over sum term can be re-expressed as $\left( \sup_{s} \sum_{\tilde{s}\in\mathcal{S}} \mathcal{G}^{\mu}_{s\tilde{s}} \right) = \sum_{\bar{a}} \mu(\bar{a}|s)\gamma^{\tau_{s\bar{a}}} \leq \gamma$, since $\tau_{s\bar{a}} \geq 1$. Thus, the operator contracts the error by at least $\gamma$ at each step. $\square$

## B. Further algorithmic details of the HSR framework

### B.1. Pseudocode for constructing HSR-NMF

---

**Algorithm S1** Computing NMF basis of expected HSR (HSR-NMF) given eigenoptions.

---

1: **Input:** MDP $\mathcal{M} = \langle \mathcal{S}, \mathcal{A}, \mathcal{P}, \mathcal{R}, \gamma \rangle$, set of pre-training tasks $G = \{g_1, \dots, g_L\}$, number of options $K$.
2: **Output:** Basis $\boldsymbol{\varphi} \in \mathbb{R}^{N\times P}$.
3: **// Stage 1: Eigenoption discovery.**
4: Compute RW-SR: $\mathbf{M}_0 \leftarrow (\mathbf{I} - \gamma\mathbf{P}_{\text{rw}})^{-1}$.
5: Compute Eigenvectors: $\mathbf{U}_0, \Sigma_0, \mathbf{V}_0 \leftarrow \text{SVD}(\mathbf{M}_0)$.
6: Construct Options set $\Omega \leftarrow \emptyset$.
7: Permute column space of $\mathbf{V}_0$ given decreasing singular values.
8: **for** $k = 1$ **to** $K$ **do**
9:     Define pseudo-reward $r_k(s, s') \leftarrow \mathbf{v}_k(s') - \mathbf{v}_k(s)$.
10:     Learn option-specific policy $\pi_k$, initial set, $\mathcal{I}_k$, and termination $\beta_k$ by maximising $r_k$ (e.g., with Q-learning).
11:     $\Omega \leftarrow \Omega \cup \{(\mathcal{I}_k, \pi_k, \beta_k)\}$.
12: **end for**
13: **// Stage 2: Offline construction of expected HSR.**
14: Initialize Expected HSR $\bar{\mathcal{M}} \leftarrow \mathbf{0}_{N\times N}$.
15: **for** each task $g \in G$ **do**
16:     Learn high-level policy $\mu_g : \mathcal{S} \to \Omega$ for task $g$.
17:     Compute *intra-option SR*: $\mathcal{B}_g \leftarrow \sum_{\bar{a}} \mu_g(\bar{a}|s)\mathbf{M}^{\bar{a}}$.
18:     Compute *continuation kernel*: $\mathcal{G}_g \leftarrow \sum_{\bar{a}} \mu_g(\bar{a}|s)\mathbf{M}^{\bar{a}}\text{diag}(\boldsymbol{\beta}_{\bar{a}})$.
19:     Solve HSR for task $g$: $\mathcal{M}_g \leftarrow (\mathbf{I} - \mathcal{G}_g)^{-1}\mathcal{B}_g$.
20:     Update Expected HSR: $\bar{\mathcal{M}} \leftarrow \bar{\mathcal{M}} + \frac{1}{M}\mathcal{L}_g$.
21: **end for**
22: **// Stage 3: Basis Discovery via NMF**
23: Solve $\min_{\Phi,H} ||\bar{\mathcal{M}} - \boldsymbol{\varphi}\mathbf{H}||_F^2$ subject to $\boldsymbol{\varphi}, \mathbf{H} \geq 0$.
24: **return** Basis features $\boldsymbol{\varphi}$.

---

### B.2. HSR with online-constructed options

To incorporate online option-construction into the HSR framework, agents are initialised with empty option sets and identity SR matrix. To construct eigenoptions, the SR matrix is updated online given primitive actions (Equation 2). Upon reaching a pre-specified number of warm-up steps, agents compute the set of eigenoptions given eigenvectors of the current SR matrix (Equation 3), which augments the action space of the agent and the associated HSR matrix can be learned hereafter (Equation 8). The option set is updated periodically instead of over each timestep to stabilise the representation learning process.

We note that the currently proposed online procedure is largely heuristic, and we leave the study of alternative procedures and associated theoretical analysis for future works.

---

**Algorithm S2** Learning HSR given online-constructed option set.

---

1: **Input:** MDP $\mathcal{M} = \langle \mathcal{S}, \mathcal{A}, \mathcal{P}, \mathcal{R}, \gamma \rangle$, number of options $K$, number of warm-up steps, $T_0$, number of steps for periodically updating the option set and associated HSR, $T_{\text{update}}$, maximum number of timesteps, $T_{\text{max}}$.
2: **Output:** HSR matrix $\mathcal{M} \in \mathbb{R}^{N \times N}$.
3: **Initialisation.** Initialise option set, $\Omega \leftarrow \emptyset$; primitive-action SR matrix, $\mathbf{M} \leftarrow \mathbf{I}_{N \times N}$, HSR matrix, $\mathcal{M} \leftarrow \mathbf{I}_{N \times N}$, policy $\pi : \mathcal{S} \times \mathcal{A} \rightarrow [0, 1]$.
4: **for** $t = 1$ **to** $T_0$ **do**
5:  Sample $a_t \sim \pi(\cdot|s)$, interact with the environment to retrieve the transition tuple $(s_t, a_t, r_t, s_{t+1})$.
6:  Update $\mathbf{M}$ given the transition tuple.
7:  Update policy with standard RL algorithms (e.g., Q-learning, policy gradient, etc.).
8: **end for**
9: Construct $\Omega$ given the current SR matrix (see lines 6-12 in Algorithm S1).
10: Augment the action space, $\bar{\mathcal{A}} = \mathcal{A} \cup \Omega$.
11: **while** $t < T_{\text{max}}$ **do**
12:  Sample $\bar{a}_t \sim \pi(\cdot|s)$, interact with the environment to retrieve the sequence of transition tuples $\{s_i, a_i, r_i, s_{i+1}\}_{i=t}^{t+\tau_{\bar{a}}}$.
13:  Update HSR (Equation 8) and SR (Equation 2) given the transition tuple sequence.
14:  **if** $(t - T_0) \mod T_{\text{update}} = 0$ **then**
15:   Update $\Omega$ given the current SR matrix.
16:  **end if**
17:  $t \leftarrow t + \tau_{\bar{a}}$.
18: **end while**
19: **return** HSR $\mathcal{M}$.

---

# C. Implementation Detail

Python codes for empirical evaluations included in the main paper can be found at https://github.com/changmin-yu/HSR_icml_2026.

## C.1. Transfer learning

All implemented agents were implemented based on Q-learning with linear function approximation. In experiments with simultaneous online learning of value function and state representation (Figure 2), all agents have access to the same set of 8 eigenoptions (Figure S1a). The same set of hyperparameters is used for training the option-specific optimal policies, online/offline construction of SR/HSR, and for learning the optimal high-level policy in downstream tasks.

- Discounting factor: $\gamma = 0.9$;

- Learning rate: $\alpha = 0.01$;

- Exploration baseline ($\epsilon$-greedy): $\epsilon = 0.1$;

- Number of SVD/NMF basis used: $K = N = 104$;

To properly assess the transferability of agents (Figure 2, Figure S2), all agents are immediately transferred to learning in the new task upon goal-switching, without re-initialisation of the linear weighting in function approximation established over the $G_1$ task. For each task, all agents are trained over 50 episodes, with a finite horizon of 5000 timesteps within each episode.

Given the augmented action space, online learning of HSR follows the derived TD-learning rule (Equation 8). We update the SR matrix also using option-induced trajectories, but with intra-option step-wise update (Algorithm S3).

Sample efficiency of learning (and of transfer) is quantified through the number of training episodes before (exponential moving average of, with step size $0.1$) episode-specific steps to reach the goal falls below $1.5\times$ optimal performance, and maximum number of episodes (50) if not converged.

For transfer learning experiments with offline-constructed predictive features (Figure 3, 4), all agents are trained over 200 episodes, with a finite horizon of 1000 timesteps within each episode.

---

**Algorithm S3** Online learning of SR matrix given option-augmented action space.

---

1: **Input:** MDP $\mathcal{M} = \langle \mathcal{S}, \mathcal{A}, \mathcal{P}, \mathcal{R}, \gamma \rangle$, set of options $\Omega = \{(\mathcal{I}_k, \pi_k, \beta_k)\}_{k=1}^{K}$, exploration baseline $\epsilon$, time horizon $T_{\max}$.
2: **Output:** SR matrix $\mathbf{M} \in \mathbb{R}^{N \times N}$.
3: Initialise SR matrix, $\mathbf{M} = \mathbf{0}_{N \times N}$, augmented action value function $Q(s, \bar{a}) = 0$ for all $(s, \bar{a}) \in \mathcal{S} \times \bar{\mathcal{A}}$, initial state $s_0$.
4: **while** $t < T_{\max}$ **and** $Q$ not converged **do**
5:     Selected action $\bar{a}_t$ via $\epsilon$-greedy given $Q$.
6:     Execute $\bar{a}_t$ until termination, yielding trajectory $\{s_t, a_t, s_{t+1}, \ldots, a_{t+\tau_{s_t \bar{a}_t}}, s_{t+\tau_{s_t \bar{a}_t}+1}\}$.
7:     **for** $\tau = 0$ **to** $\tau_{s_t \bar{a}_t}$ **do**
8:         Given single-step transition tuple $(s_{t+\tau}, a_{t+\tau}, r_{t+\tau}, s_{t+\tau+1})$
9:         Update $\mathbf{M}_{s_{t+\tau}:}$ via TD-learning (Equation 2).
10:         Update Q-function with single-step transition tuple,

$$Q(s_{t+\tau}, a_{t+\tau}) \leftarrow Q(s_{t+\tau}, a_{t+\tau}) + \alpha \left( r_{t+\tau} + \gamma \max_{\bar{a} \in \bar{\mathcal{A}}} Q(s_{t+\tau+1}, \bar{a}) - Q(s_{t+\tau}, a_{t+\tau}) \right),$$

11:     **end for**
12:     Increment step counter $t \leftarrow t + \tau_{s_t \bar{a}_t}$.
13: **end while**
14: **return** SR matrix $\mathbf{M}$.

---

## C.2. HSR and Generalised Policy Improvement

The SR-GPI algorithms, by design, require knowledge of the ground-truth reward vector for downstream tasks before any interaction with them. To ensure maximally fair comparison with the HSR agents, SR-GPI agents are initialised with linear Q-learning, and switch to the GPI policy (given max-composition of pretrained policies) once the state-dependent reward vector is constructed. Note that for simplicity, we assume that there is always one and only one goal location with non-zero reward, hence SR-GPI agents switch to GPI policies upon first encounter of the designated goal location.

## C.3. Intrinsically motivated exploration

We evaluate SR/HSR-based intrinsic exploration bonus on procedurally generated $W \times L$ grid-world environments (Figure 6a). To ensure topological diversity while ensuring existence of valid solutions, we implement a custom maze generator based on the recursive backtracking algorithm (following existing implementations from the *Minigrid* library; Chevalier-Boisvert et al. 2023). The generation process initiates with a grid fully occluded by walls. A randomised depth-first search algorithm is performed on the odd-numbered grid-coordinates: starting from a seed-specific randomly selected cell, the algorithm recursively visits unvisited neighbours, carving the intervening wall to create passages.

The reward agents observe at each timestep is a linear summation of extrinsic and intrinsic rewards.

$$\tilde{r}(s_t) = \mathcal{R}(s_t) + \lambda r_{\text{intrinsic}}(s_t, s_{t+1}), \tag{S7}$$

where we use SR/HSR-based intrinsic rewards defined in Equation 10 (Machado et al., 2020; Yu et al., 2023).

The scaling factor, $\lambda$ is set to 1 for all agents in the pure exploration task (where extrinsic rewards are always 0; Figure 6b, left; Figure 6c), and to 0.01 in the sparse-reward goal-oriented navigation setting (extrinsic reward equals 1 for all transitions into the goal state, and 0 otherwise; Figure 6b, right). All agents were implemented as canonical SARSA agents (Sutton & Barto, 1998), with $\gamma = 0.99$ and $\alpha = 0.01$, for both types of tasks.

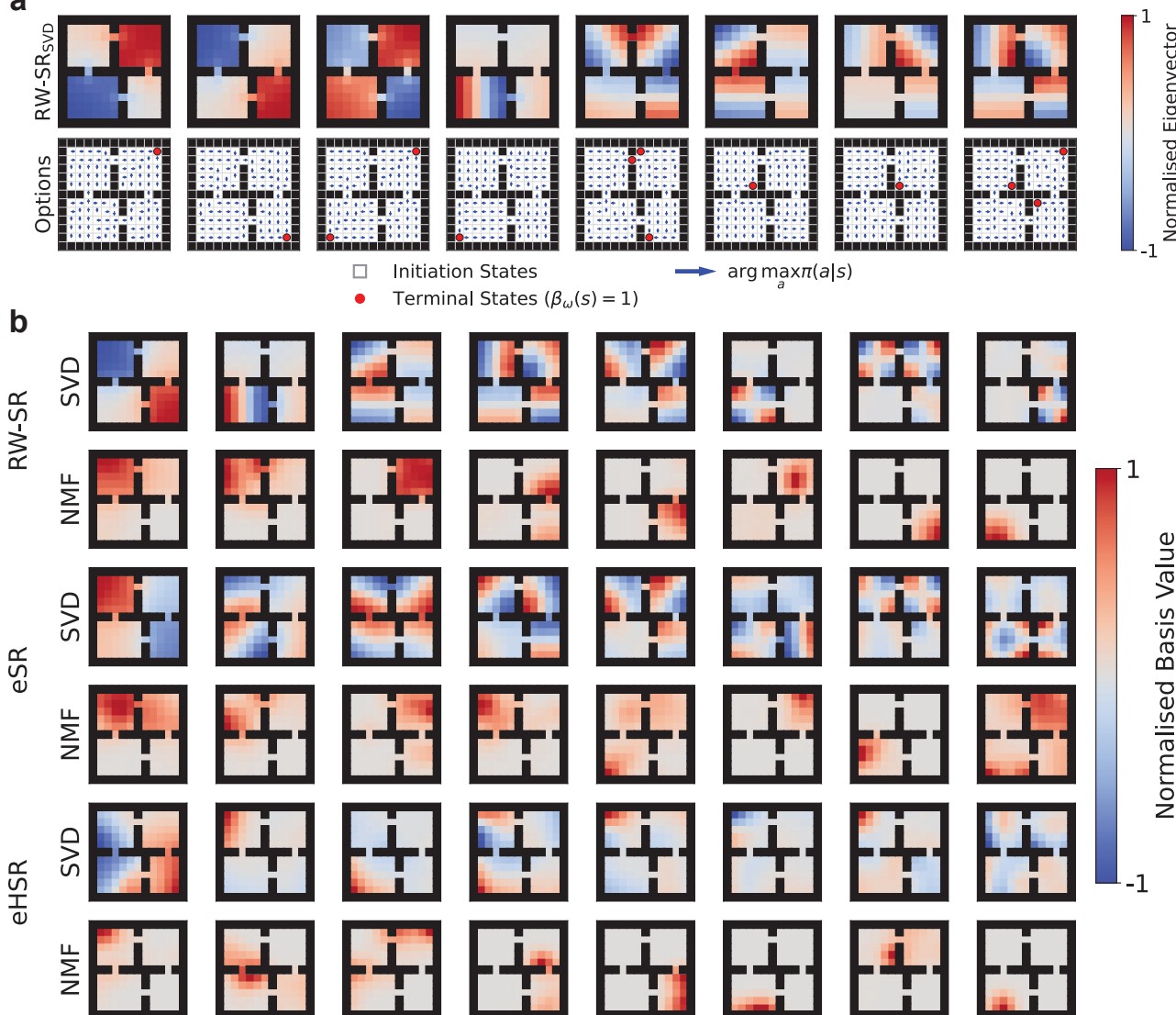

*Figure S1.* **Further empirical results on HSR-NMF. a.** The full set of (8) eigenoptions used in main experiments, both for the option-based RL agents (Figure 2) and for offline construction of expected SR/HSR (Figure 3, 4). **b.** Exemplar features given different decompositions (SVD and NMF) of RW-SR, expected SR, and expected HSR. Notably, we observe sparse HSR-NMF basis with over-representation of bottleneck states (bottom row).

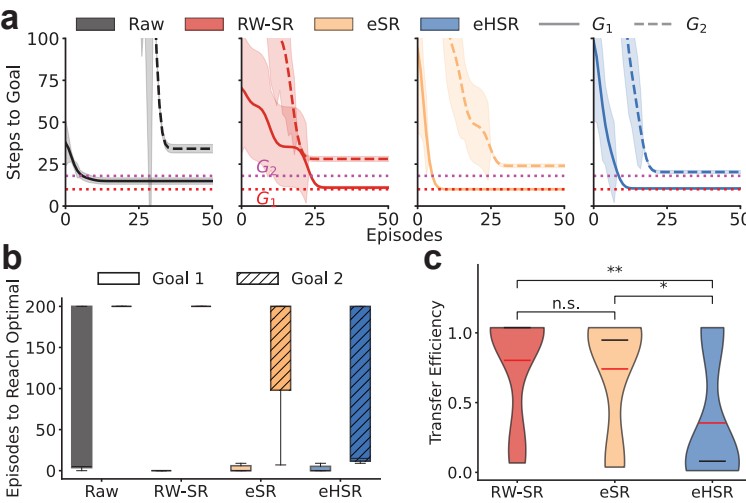

*Figure S2.* **Offline-constructed HSR facilitates more sample-efficient transfer in eigenoption-augmented agents. a.** Training curves (number of steps before reaching the goal; mean $\pm$ s.e.) of Q-learning agents with linear function approximation given row features of different offline-constructed predictive representations. All agents were first trained to reach $G_1$, then immediately transfered to learn to reach $G_2$. **b.** Number of training episodes to reach optimal performance for all agents in **a**. **c.** Transfer efficiency between $G_1$ and $G_2$ tasks for all agents.

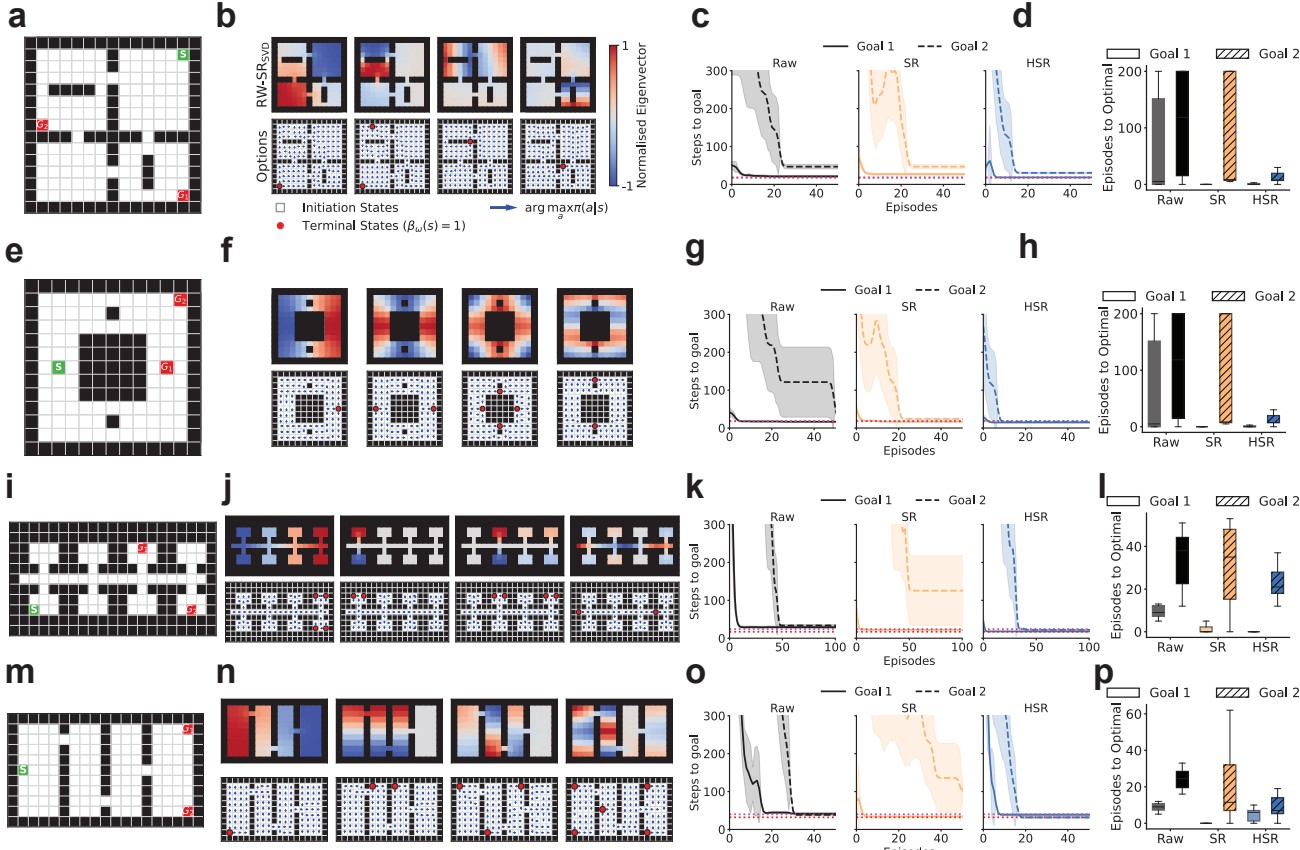

*Figure S3.* **HSR transfer-gain is persistent across multiple grid-world topologies beyond the canonical four-room environment.** **a.** Graphical demonstration of the augmented four-room environment with inserted barriers (breaking the intra-room symmetry), and location of the start (S) and the goal locations for the source ($G_1$) and transfer ($G_2$) tasks. **b.** Eigenvectors and option-specific policies for selected eigenoptions in the augmented four-room environment (cf. Figure 1d in the main text) with increasing spatial frequencies (decreasing eigenvalues). **c.** Training curves (number of primitive actions to reach the goal location as a function of number of training episodes; mean $\pm$ s.e.; 10 random seeds) for Q-learning agents with linear function approximation, given different state representations (left: one-hot representation; middle: rows of SR matrix; right: rows of HSR matrix). Note that SR- and HSR-based state representations are also trained online (cf. Figure 2b in the main text). Dashed horizontal red and magenta lines indicate the optimal number of primitive actions to reach $G_1$ and $G_2$ from S, respectively. **d.** Number of training episodes to reach optimal performance in the $G_1$ and $G_2$ tasks for all agents. **e-h.** Same as panels **a-d**, but for the "Loop" maze. **i-l.** Same as panels **a-d**, but for the "Corridor" maze. **m-p.** Same as panels **a-d**, but for the "Multi-pairwise-bottleneck" maze. Across all tested grid-world environments, HSR retains the strongest transfer performance.

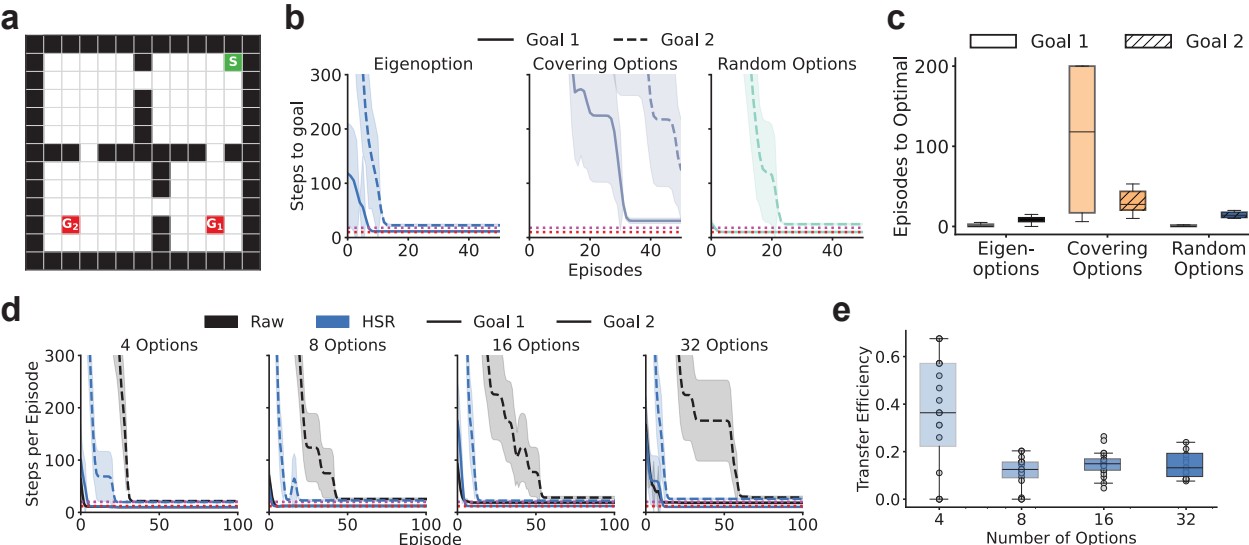

*Figure S4.* **HSR transfer-gain remains robust across option classes and largely stabilises beyond a modest option budget. a.** Graphical demonstration of the transfer learning problem setup in the four-room environment (cf. Figure 2a in the main text). **b.** Training curves (number of primitive actions to reach the goal location as a function of number of training episodes; mean ± s.e.; 10 random seeds) for Q-learning agents with linear function approximation, given online-updated HSR state representations. In addition to eigenoptions (left), we also implement agents with covering options (middle; Jinnai et al. 2019) and randomly generated target-oriented options (given randomly sampled option-specific target locations; right). HSR remains effective across alternative option classes, with eigenoptions performing best overall, random options slightly worse, and covering options substantially weaker in this transfer setting. Covering options performed poorly in our transfer setting because they are sparse-reward point options with singleton initiation sets, and are therefore only available in very limited parts of the state space. This makes them much harder to exploit as reusable transfer primitives than eigenoptions, which are broadly available and learned from shaped pseudo-rewards. This interpretation is consistent with prior analysis showing that covering options are rarely sampled online and can suffer exploration difficulties during policy learning (see also Machado et al., 2023). **c.** Number of training episodes requires to reach optimal performance in the source and target tasks for HSR agents given the three option classes. **d.** Learning curves for Raw (one-hot state representation) and HSR agents given different number of eigenoptions (4, 8, 16, or 32). Increasing the number of options enlarges the augmented action space and markedly impairs the performance of the Raw baseline, whereas HSR remains comparatively robust once a modest option budget is available. **e.** Normalised transfer efficiency (cf. Figure 2d in the main text) for HSR agents across option budgets. Transfer efficiency improves sharply by increasing the number of options from 4 to 8, and then remains broadly stable from 8 to 32 options. This suggests that beyond a small minimum set of useful options, HSR is robust to the number of incorporated eigenoptions.

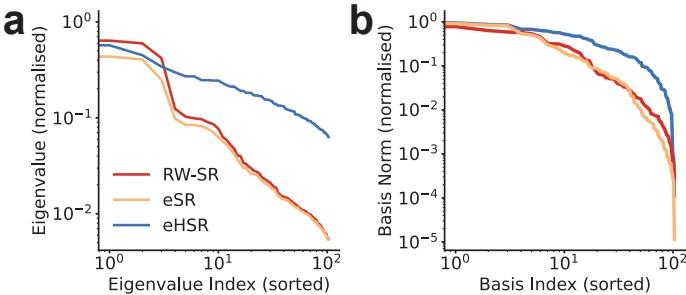

*Figure S5.* **Low-dimensional basis of HSR is heavy-tailed distributed. a.** Distribution of eigenvalues of RW-SR, eSR, and eHSR. **b.** Distribution $L_2$-norm of NMF basis of RW-SR, eSR, and eHSR.

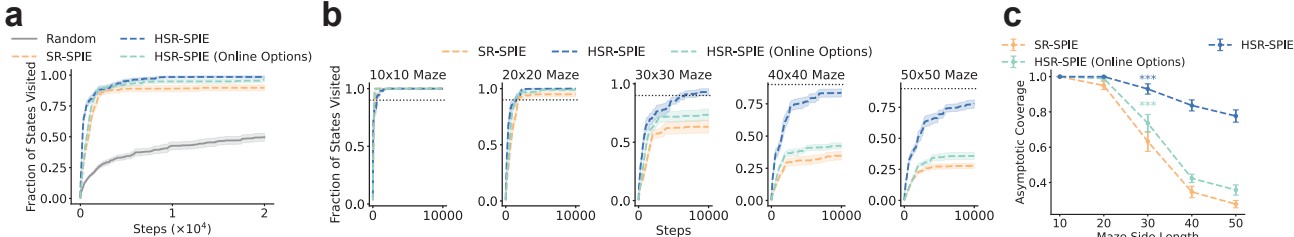

*Figure S6.* **HSR with online-constructed options facilitates stronger exploration. a.** Learning curves (mean $\pm$ s.e. over 10 random seeds) of different agents under pure exploration setting (in the absence of extrinsic reward). In addition to the agents considered in the main text (cf. Figure 6), we additionally implement an agent with HSR-based SPIE intrinsic reward, given online-constructed options (see Appendix B.2 and Algorithm S2). The HSR-SPIE agent with online-constructed options yields comparable exploration efficiency as the HSR-SPIE agent with pre-configured eigenoptions, in the $22 \times 22$ maze (Figure 6a). **b.** Same as **a**, but for mazes with increasing sizes. **c.** Asymptotic coverage (after $10,000$ primitive actions) of different agents under increasing maze sizes. Both HSR-SPIE agents with pre-computed eigenoptions and online-constructed options consistently yield stronger exploration efficiency over SR-SPIE agent across different maze sizes (two-way ANOVA test examining the effects of algorithm choice and maze size on asymptotic coverage, quantified through the interaction term between algorithm choice and maze size; HSR-SPIE vs. SR-SPIE: $p = 8.7071 \times 1^{-18}$, $F_4 = 35.5288$; HSR-SPIE with online-constructed options vs. SR-SPIE: $p = 1.4480 \times 10^{-12}$, $F_4 = 21.7160$). Overall, we found that the HSR-SPIE intrinsic reward, even under online-constructed eigenoptions, consistently outperform SR-SPIE (Yu et al., 2023). Note that despite the performance gain of HSR-SPIE (relative to SR-SPIE) given online-constructed eigenoptions is less salient comparing to its counterpart with offline-constructed eigenoptions, it does not rely on the two-stage process that requires robust pre-exploration stage, hence is more viable in practice.

