# OpenReview forum: "Hierarchical Successor Representation for Robust Transfer"
_ICML.cc/2026/Conference — ICML 2026 spotlight_

### Official Review · Reviewer_vGes · 2026-02-17

**Soundness:** 2
**Presentation:** 2
**Significance:** 2
**Originality:** 3
**Overall Recommendation:** 4
**Confidence:** 4

**Summary:**

This paper introduces the hierarchical SR (HSR), which is computed by additionally incorporating eigenoptions of the random-walk SR into the action space. The authors derive the Bellman equation for the HSR, and prove that the Bellman operator is a contraction mapping. Empirically, the authors demonstrate the following: 1) the HSR is more stable and robust as state features for value function approximation as the task (reward function) changes, 2) non-negative matrix factorization (NMF) is more suitable for the HSR than SVD, and features obtained this way perform better transfer than the SR, and 3) HSR enables better and more scalable intrinsic exploration than the SR.

**Compliance With Llm Reviewing Policy:**

Affirmed.

**Final Justification:**

I provided an initial rating of 3, due to major concerns about the soundness and clarity of the theory, and the practical utility of the proposed method (significance). The authors have been able to address these concerns with their rebuttal. I thus increase my score to 4.

**Key Questions For Authors:**

See weaknesses above. Furthermore:
1. In Section 4.2, you first train optimal policies for a few pre-training tasks, and then compute the SR and HSR for these optimal policies. Then you average these to get eSR (expected SR) and eHSR, and apply SVD/NMF to get features for learning new tasks to gauge the transfer ability of these methods. Wouldn't SR with generalized policy improvement (see Barreto et al., NeurIPS 2017) be also an interesting baseline to include, especially since you already compute the SR for optimal policies of multiple pre-training tasks?
2. It seems to me that the difference between SR and HSR, when they are both estimated using transitions generated by the high-level policy (with options as actions as well), is similar to the difference between one-step and n-step TD. Is this interpretation correct? Could the results we observe in Section 4.1 (that the HSR adapts to new tasks faster) be attributed to this?


Minor clarifying questions:
1. "Whilst SVD can compress the HSR, its orthogonal basis vectors inevitably “smear” the multi-scale structure of these manifolds across global components, leading to ringing artefacts (Gibbs phenomenon; Figure 1i; Stephane (1999)) and heavier-tailed spectral components (Figure S3a)." Can you elaborate more on this sentence? How does Figure 1i demonstrate Gibbs phenomenon?
2. In the transfer experiments (Section 4.1), can SR eventually learn to reach $G\_2$ optimally (beyond 200 episodes)?
3. Line 267: "We attribute this robustness to the stability of the underlying representation, arising from the temporal extended components in constructing the HSR that decouples predictive representations from transient variations in low-level control." Isn't decouple too strong of a word, because the HSR still has the atomic actions, unless you show that the high-level policy never selects any atomic actions? Wouldn't "alleviate the effects of transient variations" or something along this line be a more accurate description?
4. Figure 2h: What is a basis for value reconstruction? Aren't you using the entire row of the matrix as state features? How is there a varying number of bases?

Typo:
- Line 63: given efficient one-step "lienar" update
- Line 259: choose actions from the augmented action space, "consists" of primitive actions

**Limitations:**

yes

**Strengths And Weaknesses:**

Strengths:
1. The paper is in general clear and well-written, with nice visuals.
2. While the idea of using options for computing the SR exists, explicitly computing the SR with respect to a high-level policy with an action space including the options is an interesting extension.
3. Empirical analysis methods are extensive, with statistical tests.

Weaknesses:
1. I might be wrong, but I think there's quite a few **mistakes in the derivation of the HSR, and the proof of Theorem 3.1**.
   1. In Appendix A.1, in Eq. S1, while the last line looks correct to me, I think there's a mistake in the second line. First, the indices of the first and second expectation seem to be off by 1. Suppose you have an atomic action with $\tau = 1$. The summation inside the first expectation now has two terms, which is not right. The upper limit of a summation is inclusive, so I think you should subtract 1. Similar problem also exists in Eq. 9. On the other hand, the second summation also does not look right. Shouldn't there be a $\gamma$ term inside the summation? Otherwise, all of the future visits to $s'$ is weighted by the same $\gamma^{\tau_{s \bar a}}$.
	2. Also in Appendix A.1, in Eq. S2, you started with $\mathbf{F}^{\bar a}_{ss'}$, a scalar, and showed that it was equal to the multiplication of two matrices. I also don't understand the step from the second to third line.
	3. Your definition of $\mathbf{F}^{\bar a}_{ss'}$ in Eq. S2 and S3 does not look equivalent to me.
	4. In S4, the variable for the second summation should be $s''$ instead of $\tilde s$.
	5. For the proof of Theorem 3.1, the notation should be $\mathcal{M}$ instead of $\mathbf{M}$. Also, I do not understand why $\sum_{\tilde{s}} \mathcal{G}\_{s \tilde{s}}$ is bounded by $\gamma$. If I understand correctly, $\mathcal{G}\_{s \tilde s}$ is the expectation of $\mathbf{F}\_{s \tilde s}$ over the policy $\mu$. But since $\mathbf{F}\_{s \tilde s}$ can be greater than 1 (see Eq. S2), wouldn't it be possible for $\mathcal{G}\_{s \tilde s}$ to also be greater than 1?
	6. Finally, is it really ok to replace the random variable denoting an option's duration with its expectation, $\tau_{s \bar a}$, in all your derivation? Can you show some steps to justify this?
   7. General comment regarding theory in the Appendix: I suggest providing a clearer and slower derivation with more intermediate steps to improve clarity.
2. **Experiment is limited to specific set-ups.** In Section 4.1, you learn the SR and HSR online, and use their rows as state features for linear function approximation. You first train the agent to reach to first goal $G\_1$, before switching to reaching $G\_2$, and keep updating the SR/HSR throughout learning. The results show that HSR can learn to reach $G\_2$ optimally faster than SR. The conclusion that the HSR performs more robust transfer is drawn purely from results of this specific set-up. The results, though clear in its current form, would be more convincing if they are performed in not just one environment, but multiple ones. Note that I am only suggesting repeating the experiment in more grid world environments, not asking for neural network results. Section 4.2 also has this issue.
3. **Question on Practical Utility of the HSR:** I'm still struggling to grasp the practical scenarios in which the HSR would be useful, since to compute the HSR, you need to first compute the SR with a random walk policy, take SVD, and get the eigenoptions. Let's use your exploration experiment (Section 4.3) as an example, you observe that the HSR gives rise to better state space coverage. However, to compute the HSR, you need to first already explore the state space well enough to compute the eigenoptions, which are necessary for HSR computation. This seems circular to me: HSR improves exploration, but relies on first already exploring the state space well enough.

---

> ### Author Rebuttal · Authors · 2026-03-29
>
> We thank the reviewer for their constructive feedback. Below we address each
> raised comment/question, with additional empirical results in the anonymous
> repository
> (\url{https://anonymous.4open.science/r/hsr_icml_rebuttal_figures-CF85}).
>
> - Derivations of HSR
>
> In addressing point 6, we agree that using the expected duration can obscure the
> derivations, we therefore state the derivation entirely in terms of the random
> termination time, $T_{s\bar{a}}$ = termination time of (pseudo-)option
> $\bar{a}$. The derivations in the manuscript are therefore invariant to this
> change.
>
> The summation index in the first terms of Equation 9 and Equation S1 should be
> from 0 to T-1. Equation S1 is missing a $\gamma^{t-T_{s\bar{a}}}$ factor in
> the second term inside the summation.
>
> The last line in Equation S2 should be
> $[M^{\bar{a}}diag(\beta_{\bar{a}})]_{ss'}$. The transition from the second to
> the third line is a direct re-expression of the sum of future occupancy
> probability in terms of the SR associated with $\bar{a}$.
>
> The first term in Equation S3 is an equivalent re-expression of the second term
> in the second line of Equation S1. By substituting S2 into the second term in
> S3, we arrive at the final compact expression.
>
> The summation index in the second term of Equation S4 should be s''.
>
> The notation in A.2 should be $\mathcal{M}$.
>
> The rightmost term in Equation S5 is further upper bounded by
> $(\sup_s\sum_{\tilde{s}}G^\mu_{s\tilde{s}})||M-M'||$, and by definition
> (Equation 6), the supremum term can then be re-expressed as
> $(\sup_s\sum_{\tilde{s}}G^\mu_{s\tilde{s}}) =
> \sum_{\bar{a}}\mu(\bar{a}|s)E[\gamma^{T_{s\bar{a}}}|s_0=s, \bar{a}] \leq
> \gamma$, since $T_{s\bar{a}} \geq 1$. Hence the operator is bounded by $\gamma$.
>
> We will significantly expand the derivations in the appendix in the revision.
>
> - Limited experimental setup
>
> We repeated Figure 2 transfer experiments in 4 additional grid-world
> environments with distinct topology
> (multi_environment_transfer_rebuttal.pdf), and HSR consistently retained
> stronger transfer efficiency.
>
> - Practical utility of HSR
>
> To address the practical concerns arising from the two-phase processing in the
> current HSR formulations, we implemented a fully online variant in which the
> option set starts empty, agents learn primitive SR and HSR online, and
> eigenoptions are refreshed periodically from the current primitive SR after a
> short warm-up. Under this purely online protocol, HSR still improves transfer
> (four_rooms_online_option_transfer_rebuttal.pdf), and exploration
> (random_maze_exploration_online_option_rebuttal.pdf) efficiency. Thus the
> empirically observed performance gains do not rely on a fixed offline
> option-construction stage. We emphasise that this was just one possible approach
> of constructing a fully online version of HSR. For example, we expect that a
> model-based alternative (in which a transition model is constructed online and
> used to construct the off-policy random-walk SR) will perform even better.
>
> - SR-GPI baseline.
>
> We included the additional SR-GPI baseline in the four-room transfer
> experiment (sr_gpi_comparison_rebuttal.pdf), and found that the canonical
> SR-GPI (with one-hot features) performs comparably with HSR-agents, while
> combining SR-GPI with HSR-NMF representations yields the strongest performance
> improvement overall. We view this as consistent with the claim that SR-GPI
> addresses transfer learning algorithmically via policy-pool reuse, whereas HSR
> improves the transferability of the representation, and they provide
> complementary benefits that can be combined to yield further improvement.
> These additional experiments and discussions will be included in the revision.
>
> - SR vs HSR update
>
> HSR update is based on sparse N-step TD: only rows corresponding to the
> (pseudo-)option's initiation state is updated. We hypothesise the sparsity plays
> an important role in supporting stronger generalisability, hence do not expect a
> dense n-step variant would yield similar transfer performance.
>
> - Minor clarifications
>
> - We agree that "Gibbs phenomenon" was too compressed and will rephrase this
> in the revision as oscillatory ringing induced by orthogonal
> low-rank reconstruction.
>
> - SR can indeed eventually reach optimal performance in the G2 task, the key gap
> lies in the sample efficiency of transfer.
>
> - We agree that "decouple" is too strong and will soften the wording accordingly.
>
> - Figure 2h studies value reconstruction from truncated SR/HSR bases, whereas
> Figure 2b-g use full rows as state features; we will make this distinction
> explicit.
>
> - All typos will be fixed in the revision.
>
> We again thank the reviewer for their insightful feedbacks, which we believe
> have made our paper significantly stronger by addressing them. We hope the
> reviewer could raise their scores accordingly if they find our responses and
> additional empirical validations satisfyingly addressed their
> questions/concerns, and we are happy to engage in further discussions given any
> additional comment.

---

> > ### Author Rebuttal · Reviewer_vGes · 2026-04-03
> >
> > Thank you for the detailed response. My concerns are largely addressed. Final follow-up:
> >
> > Can the authors provide an example of the expanded derivation (i.e., with more steps) with the random termination time $T_{s \bar a}$?
> >
> > Doing so for Eq. S1 would be sufficient here. Thank you.

---

> > > ### Author Response · Authors · 2026-04-04
> > >
> > > We thank the reviewer for acknowledging our responses, we are glad to see that most concerns have been addressed.
> > >
> > > Regarding the followup question, we provide an expanded derivation of the Bellman recursion (original equation S1) below.
> > >
> > > $$\mathcal{M}^\mu_{ss'}=E_\mu\left[\sum_{t=0}^{\infty}\gamma^t \mathbf{1}(s_t,s')\,\middle|\, s_0=s\right]$$
> > >
> > > $$=\sum_{\bar a\in \bar{\mathcal A}}\mu(\bar a\mid s)\,E_{\bar a}\left[\sum_{t=0}^{T_{s\bar a}-1}\gamma^t \mathbf{1}(s_t,s')+\gamma^{T_{s\bar a}}\sum_{k=0}^{\infty}\gamma^k \mathbf{1}(s_{T_{s\bar a}+k},s')\,\middle|\, s_0=s,\bar{a}\right]$$
> > >
> > > $=\sum_{\bar a\in \bar{\mathcal A}}\mu(\bar a\mid s)\,E_{\bar a}\left[\sum_{t=0}^{T_{s\bar a}-1}\gamma^t \mathbf{1}(s_t,s')\,\middle|\, s_0=s,\bar a\right]+\sum_{\bar a\in \bar{\mathcal A}}\mu(\bar a\mid s)\sum_{\tilde s\in \mathcal S}E_{\bar a}\left[\gamma^{T_{s\bar a}} \mathbf{1}(s_{T_{s\bar a}},\tilde s)\,\middle|\, s_0=s,\bar a\right]\mathcal{M}^\mu_{\tilde s s'}$ [Conditioning on $s_{T_{s \bar a}} = \tilde s$.]
> > >
> > > $= \sum_{\bar a \in \bar{\mathcal A}} \mu(\bar a \mid s)\,\mathbf{M}^{\bar a}_{ss'}+$ [Apologies for the line break here since the markdown wouldn't render properly otherwise.]
> > >
> > > $\sum_{\bar{a} \in \bar{\mathcal A}}\mu(\bar{a} \mid s)\sum_{\tilde{s} \in \mathcal{S}} \mathbf{F}^{\bar a}_{s \tilde{s}}\mathcal{M}^\mu _{\tilde s s'}$
> > >
> > > $$=\mathcal{B}^\mu_{ss'}+\sum_{\tilde s\in \mathcal S}\mathcal{G}^\mu_{s\tilde s}\,\mathcal{M}^\mu_{\tilde s s'}.$$
> > >
> > > where
> > >
> > > $\mathbf{M}^{\bar a}\_{ss'}:=E_{\bar a}\left[\sum_{t=0}^{T_{s\bar a}-1}\gamma^t \mathbf{1}(s_t, s')\,\middle|\,s_0=s,\bar a\right],$ [Intra-option SR]
> > >
> > > $\mathbf{F}^{\bar a}\_{s\tilde s}:=E_{\bar a}\left[\gamma^{T_{s\bar a}} \mathbf{1}(s_{T_{s\bar a}},\tilde s)\,\middle|\, s_0=s,\bar a\right].$ [Discounted termination kernel]
> > >
> > > and we have leveraged the definitions in Equation 6 of the main paper.
> > >
> > > The expectation with respect to $\bar{a}$ is given by the trajectory generated by the option-specific policy, $\pi_{\bar{a}}$, and (stochastic) termination governed by $\beta_{\bar{a}}$. In our current simulations, all environments are deterministic, and each eigenoption terminate at a unique  terminal state, hence $T_{s\bar{a}}$ and $s_{T_{s \bar a}}$ are effectively determined by the realised option-specific trajectory. We note, however, that the derivation above does not rely on this special case.
> > >
> > > We thank the reviewer again for their acknowledgement of our rebuttal and for the followup question. We hope these have sufficiently addressed their concerns, and the reviewer could raise their scores accordingly. We are happy to engage in further discussions given any remaining concern.

---

### Official Review · Reviewer_KV5d · 2026-03-11

**Soundness:** 3
**Presentation:** 2
**Significance:** 3
**Originality:** 3
**Overall Recommendation:** 5
**Confidence:** 4

**Summary:**

The paper introduces a hierarchical extension of the successor representation (SR) framework by incorporating temporal abstractions through options. The authors argue that hierarchical goal decomposition leads to state representations that are more robust to policy changes, enabling stronger generalisation than standard SR-based reinforcement learning approaches. Furthermore, by applying non-negative matrix factorisation (NMF) to the hierarchical successor representation (HSR), the authors claim that the method gives more interpretable features and supports more efficient exploration in procedurally generated environments.

**Compliance With Llm Reviewing Policy:**

Affirmed.

**Key Questions For Authors:**

Could the authors include a brief high level description of the setting in the abstract?
Could the authors include more environments and comment on whether they have experimented with applying HSR to continuous environments?
Could the authors provide additional qualitative examples or a case study illustrating the learned options?
Could the authors include more ablations and/or discussions on the main design choices?

**Limitations:**

yes

**Strengths And Weaknesses:**

Strengths:
* Clear motivation and interesting idea: The paper clearly identifies limitations of classical SRs and proposes an interesting solution based on temporal abstraction
* Strong theoretical and mathematical formulation
* Comprehensive empirical evaluation: the experimental results test the method from multiple perspectives, including transferability, interpretability, and scalability of exploration


Weaknesses:
* I believe abstract may be difficult to follow for readers not already familiar with successor representations. A brief high-level description could improve clarity.
* Empirical review restricted to tabular reinforcement learning: All experiments are conducted in a discrete grid-world environment. It is unclear how well HSR can scale to other environments (including continuous ones).


Overall i feel that while limited on the experimental cover this is a good foundational work that the community would like to build on

---

> ### Author Rebuttal · Authors · 2026-03-29
>
> We thank the reviewer for their overall recognition of our work and for their
> insightful feedback. Below we address each
> raised comment/question, with additional empirical results in the anonymous
> repository
> (https://anonymous.4open.science/r/hsr_icml_rebuttal_figures-CF85).
>
> - Issues with abstract
>
> We agree that the current abstract presumes prior knowledge on SR-related
> concepts and moves too quickly. In the revision, we will start with a brief
> high-level problem statement up front. We provide a concrete example below
> (placed towards the end of the abstract).
>
> At a high level, the paper addresses a limitation of standard the SR: they
> support transfer well when the optimal policy remains similar, but can become
> brittle when task changes induce substantial policy changes. Our key idea is
> that incorporating temporally extended actions ("options") into the predictive
> representations, the resultant hierarchical successor representation encodes
> generalisable structure across such policy shifts. We then study how this
> representation supports three related problems: transfer learning,
> generalisable representation learning, and intrinsic exploration.
>
> - More environments and qualitative examples of options
>
> We repeated Figure 2 (in the main text) transfer experiments in 4 additional
> grid-world environments with distinct topology
> (multi\_environment\_transfer\_rebuttal.pdf), and HSR consistently retained
> stronger transfer efficiency. We additionally include qualitative demonstration
> of exemplar eigenoptions in corresponding environments.
>
> - Extending HSR to continuous environments
>
> We recognise that our current approach does not immediately generalise to MDPs
> with high-dimensional, continuous state- and action-spaces. Such extension will
> inevitably incur neural-network-based function approximation, and the present
> paper intentionally focuses on tractable settings to isolate the
> representational properties of HSR without the additional confounds of
> non-linear function approximation. That said, we view scaling to
> high-dimensional/continuous MDPs as an important future direction. The most
> natural extension is through the deep successor-feature framework, which
> naturally generalises the SR to the continuous domain. The eigenoptions
> discovery is less trivial. One viable path is to follow the heuristics proposed
> in Machado et al., 2018, where the "eigenvectors" are taken with respect to the
> matrix with the state-dependent (e.g., with uniform sampling) SFs stored as
> rows, which are then used for defining eigenoptions.
>
> We will state the scope and limitation explicitly in the revision, and to
> clarify that the present paper should be read as a foundational analysis of the
> HSR principle in tractable settings, rather than as a complete deep RL solution.
>
> - More ablations
>
> We have performed two additional ablations during the rebuttal.
>
> (a) Ablations with respect to different options.
>
> We evaluated the performance HSR-based agents given different option definitions
> (four\_room\_option\_ablation\_rebuttal.pdf), under the online transfer-learning
> setup (Figure 2 in the main text). We found that HSR-agents remains effective
> across alternative option classes, with eigenoptions performing best overall,
> random options (that preserve the length of each option) slightly worse, and
> covering options (Jinnai et al., 2019) substantially weaker.
>
> We additionally varied the number of eigenoptions used for constructing the HSR,
> and found that beyond a small minmum set of options (budget), HSR-agents are
> robust to the number of eigenoptions used in constructing the predictive
> representations.
>
> (b) HSR with online constructed options
>
> To address the practical concerns arising from the two-phase processing in the
> current HSR formulations, we implemented a fully online variant in which the
> option set starts empty, agents learn primitive SR and HSR online, and
> eigenoptions are refreshed periodically from the current primitive SR after a
> short warm-up. Under this purely online protocol, HSR still improves transfer
> (four\_rooms\_online\_option\_transfer\_rebuttal.pdf), and exploration
> (random\_maze\_exploration\_online\_option\_rebuttal.pdf) efficiency. Thus the
> empirically observed performance gains do not rely on a fixed offline
> option-construction stage. We emphasise that this was just one possible approach
> of constructing a fully online version of HSR. For example, we expect that a
> model-based alternative (in which a transition model is constructed online and
> used to construct the off-policy random-walk SR) will perform even better.
>
> We again thank the reviewer for their insightful feedback, which
> we believe have made our paper significantly stronger by addressing them. We
> hope the reviewer could raise their scores accordingly if they find our
> responses and additional empirical validations satisfyingly addressed their
> questions/concerns, and we are happy to engage in further discussions given any
> additional comment.

---

> > ### Author Rebuttal · Reviewer_KV5d · 2026-04-03
> >
> > I want to thank the authors for their detailed response. The changes will definitely make the paper a better foundational work the community can extend to continuous settings in the future through deep learning representation.
> >
> > I maintain my acceptance score.

---

> > > ### Author Response · Authors · 2026-04-04
> > >
> > > We thank the reviewer for acknowledging our responses, we are glad to see all concerns have been addressed.

---

### Official Review · Reviewer_dxaw · 2026-03-13

**Soundness:** 4
**Presentation:** 4
**Significance:** 3
**Originality:** 3
**Overall Recommendation:** 5
**Confidence:** 4

**Summary:**

This paper introduces hierarchical successor representation (HSR) as a method for robust cross-task transfer. While standard SRs are defined as expected discounted future occupancy under some policy $\pi: \mathcal S \rightarrow \mathcal A$, HSRs are expected discounted future occupancy under some high-level policy $\mu: \mathcal S \rightarrow \bar{\mathcal{A}}$, where $\bar{\mathcal{A}}$ is a union of low-level actions $\mathcal A$ and high-level options $\Omega$. The options considered in this paper are derived from the eigenvectors of the random walk successor representations (RW-SR) [1]. Having defined HSR, the paper proceeds to derive a Bellman operator for HSR and prove that it is a contraction. Hence, HSRs can be learned via iterative Bellman updates. This induces a Q-learning algorithm that jointly learns an optimal policy and an HSR, applicable to online sequential task transfer. Next, the paper proposes to use non-negative matrix factorisation (NMF) to extract basis vectors from HSR, which can be used as representations for sample-efficient transfer to new tasks. Since the downstream task is unknown during pretraining, the representations are extracted from the expected HSR (eHSR) of multiple pretrained policies. Finally, the paper proposes an efficient exploration method with HSR as an intrinsic reward. To summarize the logical flow, we have

RW-SR -> Eigen options -> HSR (transfer, exploration) -> eHSR -> NMF (representation)

The paper validates the claims with three sets of experiments on a tabular maze environment.
1. (Transfer) HSR improves generalization, transferring from task 1 to task 2 faster than standard SR.
2. (Representation) low-rank decomposition of eHSR via NMF leads to sparse, interpretable representations, which enable more efficient transfer to downstream tasks than standard SR + NMF.
3. (Exploration) HSR enables more efficient exploration compared to standard SR (using either SR-NORM or SPIE for intrinsic rewards).

Overall, the paper advances the successor representation line of work by proposing a hierarchical formulation that improves robustness and transfer efficiency.

[1] Marlos C. Machado, Clemens Rosenbaum, Xiaoxiao Guo, Miao Liu, Gerald Tesauro, Murray Campbell. Eigenoption Discovery through the Deep Successor Representation. 2017.

**Compliance With Llm Reviewing Policy:**

Affirmed.

**Final Justification:**

My original evaluation was positive and the rebuttal clarified all the conceptual questions I had. I maintain my score.

**Key Questions For Authors:**

1. What is the Q-learning algorithm used for the representation learning (HSR-NMF) experiments?
2. How do you see the method extending to high-dimensional and/or continuous settings? Can the same mechanism be extended to successor features?
3. What happens if you use a different set of options? It would be helpful to have this ablation to isolate the gains from the option design and the algorithm.

Typos:
1. Line 151 left: "Despite the SR framework provides" -> providing.

**Limitations:**

Yes.

**Strengths And Weaknesses:**

**Strengths**
1. The proposed method, HSR, makes a valuable contribution to the successor representation line of work by improving the robustness and transferability.
2. The claims are backed up by strong empirical results. The paper shows that HSR adapts faster than SR, induces transferable representations, and enables efficient exploration.
3. The presentation of the paper is excellent. The method and experiment sections are written rigorously. The figures are remarkably clear and informative. For example, Figure 1. e, f, and Figure 2. e, f clearly present the difference between SR and HSR: the former is overlapping and dissipative, whereas the latter is sparse and concentrated.

**Weaknesses**
1. The method is relatively complicated since it requires defining a set of options first.
2. The eigen options and matrix-form SR considered in the paper are only applicable in discrete, tabular environments. It is unclear how well the method applies to continuous or high-dimensional settings.

---

> ### Author Rebuttal · Authors · 2026-03-29
>
> We thank the reviewer for their overall recognition of our work and for their
> insightful feedback. Below we address each
> raised comment/question, with additional empirical results in the anonymous
> repository
> (https://anonymous.4open.science/r/hsr_icml_rebuttal_figures-CF85).
>
> - HSR with online discovered options
>
> To address the practical concerns arising from the two-phase processing in the
> current HSR formulations, we implemented a fully online variant in which the
> option set starts empty, agents learn primitive SR and HSR online, and
> eigenoptions are refreshed periodically from the current primitive SR after a
> short warm-up. Under this purely online protocol, HSR still improves transfer
> (four\_rooms\_online\_option\_transfer\_rebuttal.pdf), and exploration
> (random\_maze\_exploration\_online\_option\_rebuttal.pdf) efficiency. Thus the
> empirically observed performance gains do not rely on a fixed offline
> option-construction stage. We emphasise that this was just one possible approach
> of constructing a fully online version of HSR. For example, we expect that a
> model-based alternative (in which a transition model is constructed online and
> used to construct the off-policy random-walk SR) will perform even better.
>
> - Extension to continuous MDPs
>
> We recognise that our current approach does not immediately generalise to MDPs
> with high-dimensional, continuous state- and action-spaces. Such extension will
> inevitably incur neural-network-based function approximation, and the present
> paper intentionally focuses on tractable settings to isolate the
> representational properties of HSR without the additional confounds of
> non-linear function approximation. That said, we view scaling to
> high-dimensional/continuous MDPs as an important future direction. The most
> natural extension is through the deep successor-feature framework, which
> naturally generalises the SR to the continuous domain. The eigenoptions
> discovery is less trivial. One viable path is to follow the heuristics proposed
> in Machado et al., 2018, where the "eigenvectors" are taken with respect to the
> matrix with the state-dependent (e.g., with uniform sampling) SFs stored as
> rows, which are then used for defining eigenoptions.
>
> We will state the scope and limitation explicitly in the revision, and to
> clarify that the present paper should be read as a foundational analysis of the
> HSR principle in tractable settings, rather than as a complete deep RL solution.
>
> - Q-learning algorithm in representation learning experiments.
>
> During the representation-learning phase, we firstly solve for the optimal
> high-level policy (given the set of eigenoptions) for each of the pre-training
> environments, with tabular Q-learning methods (treating each option as a valid
> action). During downstream transfer learning tasks, we train Q-learning
> algorithms with linear function approximation given different representations
> (including the HSR-NMF, see, e.g., Algorithm S1 in the appendix).
>
> - Robustness to different options.
>
> We hypothesise that the utility of the HSR framework mainly arises from both its
> sparse representation (semi-Markovian Bellman updates) and the geometry
> awareness (given eigenoptions). Hence, we expect changing the set of options
> used for constructing the HSRs will retain the benefits provided by the sparse
> updates, but the resulting performance might differentially change dependent on
> the geometry information inherent in the options used. To this end, we performed
> an additional experiment by evaluating the performance HSR-based agents given
> different option definitions (four\_room\_option\_ablation\_rebuttal.pdf), under
> the online transfer-learning setup (Figure 2 in the main text). We found that
> HSR-agents remains effective across alternative option classes, with
> eigenoptions performing best overall, random options (that preserve the length
> of each option) slightly worse, and covering options (Jinnai et al., 2019)
> substantially weaker. The covering options, despite providing the geometry
> information, are point options with singleton initiation sets, and are therefore
> only available in very limited parts of the state space. This hence makes them
> much harder to exploit the reusable transfer primitives than eigenoptions or
> random options (see also Machado et al., 2023).
>
> We additionally evaluated the transfer efficiency of HSR-agents with varying
> number of eigenoptions, and found that beyond a small minmum set of options
> (budget), HSR-agents are robust to the number of eigenoptions used in
> constructing the predictive representations.
>
> - All typos will be fixed in the revision.
>
> We again thank the reviewer for their insightful feedback, which we believe have
> made our paper significantly stronger by addressing them. We hope the reviewer
> could raise their scores accordingly if they find our responses and additional
> empirical validations satisfyingly addressed their questions/concerns, and we
> are happy to engage in further discussions given any additional comment.

---

> > ### Author Rebuttal · Reviewer_dxaw · 2026-04-03
> >
> > Thank you for the detailed clarifications. All my questions were resolved, and I maintain my evaluation of the work.

---

> > > ### Author Response · Authors · 2026-04-04
> > >
> > > We thank the reviewer for acknowledging our responses, we are glad to see all concerns have been addressed.

---

### Official Review · Reviewer_Udwi · 2026-03-14

**Soundness:** 2
**Presentation:** 2
**Significance:** 3
**Originality:** 3
**Overall Recommendation:** 4
**Confidence:** 4

**Summary:**

The paper proposes moving beyond successor features defined over flat policies that select primitive actions, and instead computes successor features over hierarchical policies that select options. The authors argue that flat successor representations suffer from spectral diffusion, which leads to dense and overlapping features that scale poorly. In contrast, hierarchical successor features mitigate this issue by operating at the level of options rather than primitive actions. The proposed methodology is evaluated on a set of procedurally generated environments.

**Compliance With Llm Reviewing Policy:**

Affirmed.

**Final Justification:**

I slightly increased my score while I still see as a major limitation the restriction of this work on simple tabular settings the additional experiments and responses from the authors have clarified my doubt and reported usefull additional comparison with Option Keyboard approaches.

**Key Questions For Authors:**

- The paper lacks a discussion and comparison with relevant previous work such the Option-Keyboard and Combining Behaviors with the Successor Features Keyboard that are related to the proposed approach. Can you please provide a discussion on how your algorithm relates and compare to this previous works?

- Results are limited to tabular discrete MDP and is not obvious how the proposed algorithm can be generalized to high dimensional continous mdps. Can you clearly discuss the limitation of the proposed approach in relation to the extension to more general settings like high dimensional observation and continous MDPs?

**Limitations:**

yes

**Strengths And Weaknesses:**

Strength:

- The idea and the problem addressed in the paper are interesting, as they connect concepts from several algorithms, eigenoptions, successor features, and non-negative matrix factorization in a unified and original way.

- The problem studied in the paper is of interest to the reinforcement learning community, particularly in relation to how knowledge can be transferred between tasks within the same environment and across similar environments.

Weaknesses:

- The paper lacks a discussion and comparison with relevant previous work such the Option-Keyboard and Combining Behaviors with the Successor Features Keyboard that are related to the proposed approach.

- The paper is hard to follow, as it moves directly into technical details without clearly and formally stating the problem it aims to solve. Providing a clearer and more complete problem statement would help readers follow the paper more easily.

- The paper is confusing in the sense that is trying to cover many setups like state representation, intrinsic motivation, exploration making it hard to follow.

- Results are limited to tabular discrete MDP and is not obvious how the proposed algorithm can be generalized to high dimensional continous mdps.

- The paper relies on a two stage procedure where the options are first learned through pre-exposure to the environment and the hierarchical successor representation is then learned in a following phase. Is not clear to me wheter the algorithm can be easily extended to the pure online setting or if the two stages procedure is actually needed to stabilize the learning.

---

> ### Author Rebuttal · Authors · 2026-03-29
>
> We thank the reviewer for their constructive feedback. Below we address each
> raised comment/question, with additional empirical results in the anonymous
> repository
> (\url{https://anonymous.4open.science/r/hsr_icml_rebuttal_figures-CF85}).
>
> - Related works: option/SF-keyboard
>
> We thank the reviewer for pointing out the close connection to the SF/GPI line
> of work, especially with SF-GPI (Barreto et al., 2016), the option keyboard
> (Barreto et al., 2019) and the more recent successor features keyboard (Carvalho
> et al., 2023), which were missing from the manuscript. These methods address
> transfer learning by storing a library of previously learned policy, and then
> composing them under a new task using GPI. Our HSR framework is related but
> distinct: rather than composing pretrained policies at transfer time, it seeks
> to construct a single predictive representation that is reusable across policy
> changes. To address this point empirically, we added an SR-GPI baseline in the
> experimental setting of Figure 3 in the main text
> (sr_gpi_comparison_rebuttal.pdf). To ensure a fair comparison, the
> ground-truth reward function in the downstream transfer task was not provided a
> priori to the SR-GPI agent, but needed to be inferred online after the first
> rewarded transition, after which the stored SRs were re-evaluated and composed
> via GPI. Under this setup, the canonical SR-GPI with a one-hot state
> representation performed comparably to the HSR agents that did not compose
> policies. Moreover, combining the SR-GPI agent with HSR state representations
> yielded even stronger overall performance, thus supporting the view that GPI and
> HSR offer complementary benefits: policy-composition on the one hand, and
> generalisable representation on the other. We will include discussions with
> respect to the missing reference and clarify the distinctions in the revision.
>
> - Issues with presentation
>
> We agree that the current presentation moves too quickly across several
> experimental settings. Our goal was not to present multiple unrelated methods,
> but show that a single object, HSR, improves three closely related problems:
> online transfer learning; generalisable representation learning; and intrinsic
> exploration. In the revision, we will make this organising principle explicit
> upfront by revising the abstract/introduction, adding a roadmap paragraph, and
> including a compact schematic/table linking the three settings.
>
> - Extension to continuous MDPs
>
> We recognise that our current approach does not immediately generalise to MDPs
> with high-dimensional, continuous state- and action-spaces. Such extension will
> inevitably incur neural-network-based function approximation, and the present
> paper intentionally focuses on tractable settings to isolate the
> representational properties of HSR without the additional confounds of
> non-linear function approximation. That said, we view scaling to
> high-dimensional/continuous MDPs as an important future direction. The most
> natural extension is through the deep successor-feature framework, which
> naturally generalises the SR to the continuous domain. The eigenoptions
> discovery is less trivial. One viable path is to follow the heuristics proposed
> in Machado et al., 2018, where the "eigenvectors" are taken with respect to the
> matrix with the state-dependent (e.g., with uniform sampling) SFs stored as
> rows, which are then used for defining eigenoptions.
>
> We will state the scope and limitation explicitly in the revision, and to
> clarify that the present paper should be read as a foundational analysis of the
> HSR principle in tractable settings, rather than as a complete deep RL solution.
>
> - HSR with online discovered options
>
> To address the practical concerns arising from the two-phase processing in the
> current HSR formulations, we implemented a fully online variant in which the
> option set starts empty, agents learn primitive SR and HSR online, and
> eigenoptions are refreshed periodically from the current primitive SR after a
> short warm-up. Under this purely online protocol, HSR still improves transfer
> (four_rooms_online_option_transfer_rebuttal.pdf), and exploration
> (random_maze_exploration_online_option_rebuttal.pdf) efficiency. Thus the
> empirically observed performance gains do not rely on a fixed offline
> option-construction stage. We emphasise that this was just one possible approach
> of constructing a fully online version of HSR. For example, we expect that a
> model-based alternative (in which a transition model is constructed online and
> used to construct the off-policy random-walk SR) will perform even better.
>
> We again thank the reviewer for their insightful feedback, which
> we believe have made our paper significantly stronger by addressing them. We
> hope the reviewer could raise their scores accordingly if they find our
> responses and additional empirical validations satisfyingly addressed their
> questions/concerns, and we are happy to engage in further discussions given any
> additional comment.

---

> > ### Author Rebuttal · Reviewer_Udwi · 2026-04-04
> >
> > I thank the reviewer for the detailed response. I'm happy with the clarification and comparison of the role of previous work such as SR-GPI and online settings results. I will consider slightly increasing my score but I still see as a major limitation the restriction of this work on simple tabular settings.

---

> > > ### Author Response · Authors · 2026-04-04
> > >
> > > Thank you for the follow-up and for acknowledging that our responses on related prior work, comparison with SR-GPI baselines, and extension to the purely online-setting addressed most concerns. We also appreciate that the remaining issue has narrowed to the tabular scope of the current paper. We fully agree that extension to high-dimensional / continuous settings is an important future direction, but our intended claim here is more modest: a foundational analysis of the HSR principle in tractable settings, not a complete solution for deep/continuous RL. Since the core technical, empirical, and positioning concerns now appear largely resolved, we hope you might reconsider adjusting the scores based on whether the remaining scope limitation should continue to outweigh the merits of the current contribution.

---

### Decision · Program_Chairs · 2026-04-30

**Decision:**

Accept (spotlight)

**Comment:**

This paper proposes an extension of the successor representation that takes into consideration temporally-extended actions. The usefulness of such a representation is demonstrated by using it (or its eigenvectors) to tackle problems such as exploration and option discovery. This can be quite a foundational result, and eventually, all reviewers agreed the paper should be accepted. Personally, I do not think some of the concerns raised in the reviews (or discussion phase) related to the paper being developed only in tabular discrete MDPs (or in it relying on a two-stage procedure for option discovery) would be grounds for rejection given that all the foundational results in the field were obtained this way, in a series of papers starting in simpler settings that allow for better understanding and control.